# HUMANOIDVERSE: A VERSATILE HUMANOID FOR VISION-LANGUAGE GUIDED MULTI-OBJECT REARRANGEMENT

## ABSTRACT

We introduce HumanoidVerse, a novel framework for vision-language guided humanoid control that enables a single physically simulated robot to perform long-horizon, multi-object rearrangement tasks across diverse scenes. Unlike prior methods that operate in fixed settings with single-object interactions, our approach supports consecutive manipulation of multiple objects, guided only by natural language instructions and egocentric camera RGB observations. HumanoidVerse is trained via a multi-stage curriculum using a dual-teacher distillation pipeline, enabling fluid transitions between sub-tasks without requiring environment resets. To support this, we construct a large-scale dataset comprising 350 multi-object tasks spanning four room layouts. Extensive experiments in the Isaac Gym simulator demonstrate that our method significantly outperforms prior state-of-the-art in both task success rate and spatial precision, and generalizes well to unseen environments and instructions. Our work represents a key step toward robust, general-purpose humanoid agents capable of executing complex, sequential tasks under real-world sensory constraints.

## 1 INTRODUCTION

Humanoid robots capable of imitating human actions, interacting with scenes, and manipulating objects have broad practical value. Existing work has advanced humanoid control (Peng et al., 2021; Tessler et al., 2023; Luo et al., 2023; Zhu et al., 2023), environment interaction (Starke et al., 2019; Hassan et al., 2023; Xiao et al., 2023; Pan et al., 2024), and object manipulation (Xu et al., 2024; Wang et al., 2023; Xie et al., 2023), yet remains limited to simple, single-object tasks (Wang et al., 2024; Xu et al., 2025; Pan et al., 2025) and often relies on full environment knowledge, unavailable in real-world settings.

Recent vision-language approaches (Xu et al., 2024; Ding et al., 2025) enable humanoid action control but still manipulate only one object from fixed states, lacking long-horizon and multi-object capabilities. Real-world use demands continuous, multi-object manipulation and real-time task transitions—capabilities absent from existing systems.

To address these limitations, we present **HumanoidVerse**, a vision-language guided humanoid policy for long-horizon, multi-object rearrangement with a single model. As shown in Figure 1, the robot receives a natural language instruction, uses ego-centric camera RGB observations to relocate the target object, then seamlessly transitions to subsequent tasks from its current position without disturbing previously placed items.

**HumanoidVerse** is, to our knowledge, the first system enabling humanoids to execute physically stable, continuous multi-object rearrangement. We train a teacher policy using the AMP framework (Peng et al., 2021) with goal-conditioned reinforcement learning, combining adversarial motion and task rewards for realistic, task-completing behaviors. The teacher learns stable transport and safe release through a multi-stage curriculum with privileged state access. We then distill it into a student VLA policy via DAgger (Ross et al., 2011), operating solely on RGB and language inputs, aided by a transition detection mechanism for robust task switching.

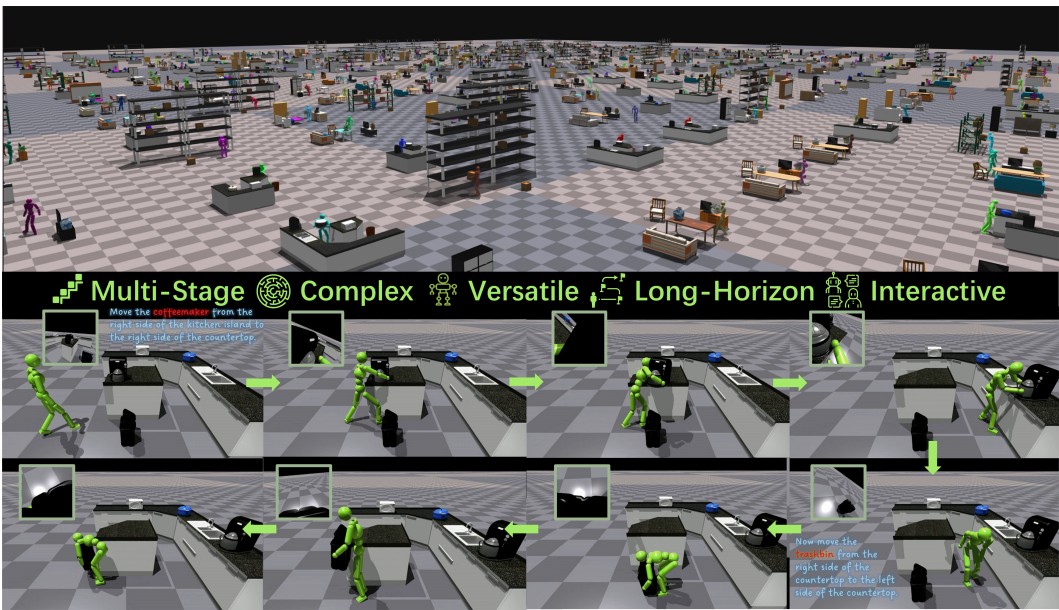

Figure 1: Top: 350 diverse tasks across four distinct room layouts. Bottom: An example of a long-horizon, multi-object rearrangement task. A human issues two consecutive natural language commands; the robot perceives via its head camera and completes each task in sequence. The process is multi-stage, interactive, and complex. A single model handles varied tasks across different scenes, demonstrating strong versatility.

To support this, we extend the HITR dataset (Xu et al., 2024) to 350 sequential two-object tasks across four layouts—bedroom, kitchen, living room, and warehouse—trained and evaluated in Isaac Gym (Makoviychuk et al., 2021). Experiments validate both the dataset and our model's ability to perform smooth, long-horizon multi-object rearrangement.

In summary, our contributions are:

- We introduce **HumanoidVerse**, the first humanoid system that integrates egocentric vision and natural language to accomplish long-horizon multi-object rearrangement tasks smoothly with real-time human interaction.
- We construct a new dataset for this task, covering 350 tasks in four distinct room layouts, and perform extensive experiments to validate the effectiveness of **HumanoidVerse**.

## 2 RELATED WORK

### 2.1 MOTION SYNTHESIS AND PHYSICALLY-BASED HUMAN-SCENE INTERACTION (HSI)

Motion synthesis spans graphics, vision, and robotics, and is commonly divided into kinematics-based (Li et al., 2023; Jiang et al., 2024) and physics-based approaches (Peng et al., 2018; 2021; Zhu et al., 2023; Tessler et al., 2023). Kinematic methods generate semantically consistent motions via generative models, while physics-based methods learn control policies in simulators for realistic, physically plausible behaviors. Representative systems such as DeepMimic (Peng et al., 2018) and AMP (Peng et al., 2021) have been extended with adversarial priors, discrete latent variables, and language-conditioned control. HSI research focuses on synthesizing interactions that satisfy both task goals and physical realism. Works like InterPhys (Hassan et al., 2023) and InterScene (Pan et al., 2024) address dynamic object interaction, while early methods relied on inverse kinematics or handcrafted controllers. Recent deep RL approaches (Wang et al., 2024; Xu et al., 2025) demonstrate complex skills such as dribbling or utensil use. Despite progress, most controllers are task-specific and handle only simple objects, lacking scalability to diverse geometries or contact-rich, multi-

object interactions. Our work aims to develop a general-purpose, robust policy for long-horizon, multi-object rearrangement in everyday scenes.

## 2.2 OBJECT REARRANGEMENT IN ROOM SCENES

Object rearrangement is a key task in embodied AI, involving navigation and manipulation. Existing benchmarks (Weihs et al., 2021; Yenamandra et al., 2023) address visual restoration or open-vocabulary pick-and-place, with some leveraging commonsense from language models (Wu et al., 2023). However, most use simplified embodiments such as mobile bases with grippers, limited to flat terrains and small objects. HumanVLA (Xu et al., 2024) extends to humanoid manipulation of heavier objects, but remains restricted to single-object, fixed-start tasks without sequential execution. Our work introduces the first benchmark for continuous, long-horizon humanoid rearrangement across multiple objects, representing the most complex setting to date.

## 2.3 VISION- AND LANGUAGE-GUIDED HUMAN-SCENE INTERACTION

Recent work has explored integrating vision and language into humanoid control. Early studies generated scene-agnostic motions from language (Chen et al., 2023; Jiang et al., 2023), while later efforts tackled language-conditioned interactions in physical environments using predefined task structures (Zhao et al., 2022) or large language/vision-language models for action generation (Zitkovich et al., 2023). Hybrid pipelines (Luo et al., 2023) improve scalability but remain tied to predefined trajectories, limiting physical realism. Vision-language-action (VLA) models (Wang et al., 2025; Driess et al., 2025) show promise but mainly target simple manipulators; humanoid-level VLA control remains nascent, with HumanVLA (Xu et al., 2024) and Humanoid-VLA (Ding et al., 2025) as early examples.

Current approaches are constrained to single-instruction episodes, limited real-time interaction, and focus on posture over rich object-scene manipulation. Our **HumanoidVerse** addresses these gaps, enabling real-time multi-stage interaction, seamless task transitions, and complex multi-object rearrangement guided by vision and language.

## 3 TRAINING PROCESS OF HUMANOIDVERSE

Figure 2 shows the training pipeline. Due to the complexity of humanoid locomotion and stability constraints, manual teleoperation is infeasible; we therefore adopt a teacher–student framework. The teacher is trained via reinforcement learning with oracle state inputs (e.g., object poses, geometry, navigation waypoints, targets), and distilled into a student model using DAgger (Ross et al., 2011). The student receives first-person RGB and language inputs, producing control actions in a vision-language-action (VLA) manner.

Given the difficulty of long-horizon, multi-object rearrangement, we employ a multi-stage curriculum. A first teacher learns single-object rearrangement, then is extended to handle post-placement behaviors—stable release, retreat, and collision avoidance. For multi-object tasks, a second teacher starts from the state left by the first, learning to operate from variable configurations and navigate around obstacles.

### 3.1 STAGE 1: PRETRAINING FOR SINGLE-OBJECT REARRANGEMENT

In prior related tasks, the robot typically starts from a fixed location in the scene and always begins in a standing posture. However, our task involves sequential multi-object rearrangement, where the robot must handle a second object after completing the first. If the robot proceeds directly from the post-placement configuration of the first object to manipulate the second, it will encounter out-of-distribution states, often leading to task failure. To mitigate this, we introduce a dedicated pretraining phase that equips the model with robust single-object rearrangement skills from diverse initial configurations.

To accelerate training and leverage existing visuomotor capabilities, we initialize our model using a pretrained checkpoint from HumanVLA Xu et al. (2024), which has been trained on diverse human

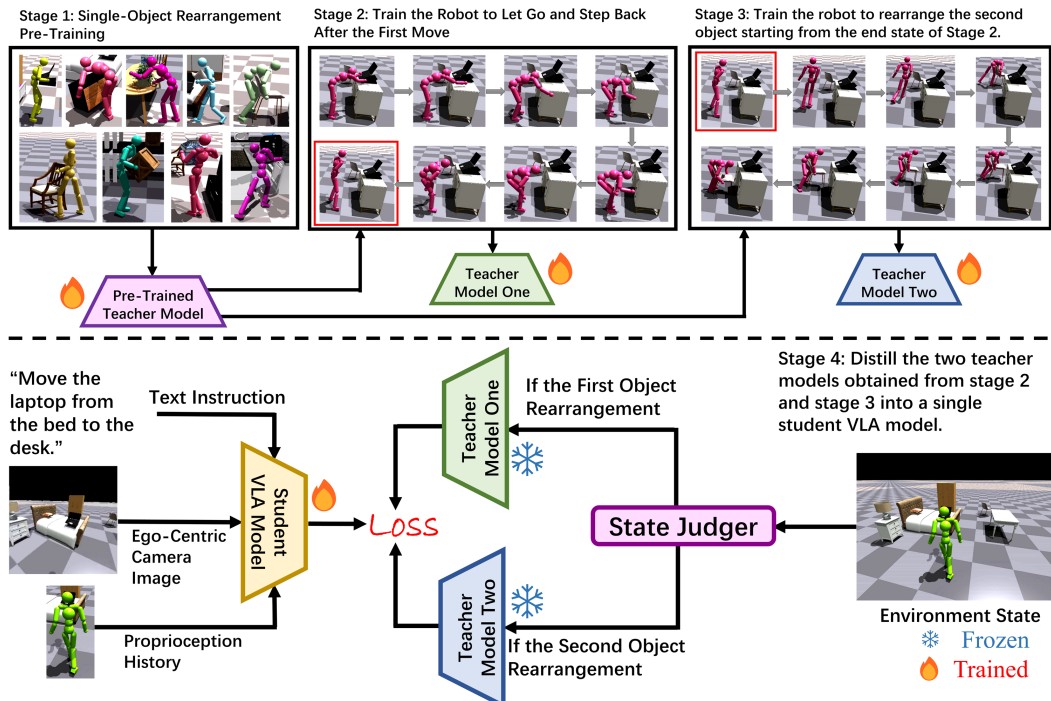

Figure 2: Multi-stage training pipeline of **HumanoidVerse**: Stage 1 trains a teacher model via RL for single-object rearrangement. Stage 2 extends it to releasing and stepping back. Stage 3 trains a second teacher model to manipulate a second object from the end state of Stage 2. In Stage 4, the student VLA model, taking real-time visual and language inputs, is distilled from the two teacher models. The first teacher model is used to distill during the initial rearrangement, and the second after releasing and retreating. Both are unified into a single student model for full multi-object rearrangement.

motion data to align vision, language, and action representations. This initialization provides a strong prior for low-level motor control and high-level grounding between language and behavior.

To construct the pretraining dataset, we decompose the 350 original tasks in our multi-object rearrangement dataset. Each original task involves manipulating two objects in sequence. As illustrated in Algorithm 1, we split each task into two separate single-object tasks: the first starts from the standard initial position and involves manipulating the first object; the second begins near the post-placement location of the first object and involves manipulating the second object. This process yields a total of 700 single-object tasks, enabling the robot to generalize over various starting states.

---

**Algorithm 1** Pretraining Task Generation from Multi-Object Dataset

---

1: **Input:** Multi-object tasks $\mathcal{T} = \{T_1, T_2, \ldots, T_{350}\}$
2: **Output:** Single-object pretraining tasks $\mathcal{T}_{\text{single}}$
3: **for** each $T_i \in \mathcal{T}$ **do**
4:     Split $T_i$ into $(O_1, O_2)$: two sequential objects
5:     Create task $T_i^{(1)}$: manipulate $O_1$ from default initial state
6:     Create task $T_i^{(2)}$: manipulate $O_2$ from post-$O_1$ configuration
7:     Add $T_i^{(1)}, T_i^{(2)}$ to $\mathcal{T}_{\text{single}}$
8: **end for**
9: **return** $\mathcal{T}_{\text{single}}$ with $|\mathcal{T}_{\text{single}}| = 700$

---

Pretraining on these 700 single-object tasks serves as a strong foundation for the subsequent multi-object training, allowing the model to handle a wide range of robot poses and spatial arrangements.

To further guide the robot toward human-like motion and object interaction, we incorporate a style reward using Adversarial Motion Prior (AMP) ( Peng et al. (2021)). The AMP reward encourages the robot to produce motion sequences that resemble those of a reference human demonstrator. It is defined as:

$$r_{\text{style}} = r^{\text{AMP}}(s_{t:t+1}) = \lambda_{\text{AMP}} \cdot [-\log(1 - D_\phi(s_{t:t+1}, a_t))] \tag{1}$$

where $D_\phi(s_{t:t+1}, a_t)$ is a learned discriminator that distinguishes between human-like and non-human-like transitions, based on consecutive states $s_{t:t+1}$ and the action $a_t$, and $\lambda_{\text{AMP}}$ is a weighting coefficient that balances the influence of the style reward. This formulation penalizes deviations from human motion patterns while allowing the robot to explore diverse yet plausible behaviors during pretraining.

The total reward used during pretraining combines task success and motion style:

$$r_{\text{total}} = r_{\text{task}} + r_{\text{style}} \tag{2}$$

where $r_{\text{task}}$ quantifies task success (see Appendix A.1 for the detailed formulation) and $r_{\text{style}}$ is defined as in Equation 1 to encourage motion style consistency.

This pretraining stage ensures the robot can execute single-object rearrangement tasks from both canonical and non-canonical starting poses, preparing it for the long-horizon demands of multi-object rearrangement.

### 3.2 STAGE 2: LEARNING TO RELEASE AND STEP BACK AFTER THE FIRST REARRANGEMENT

Building on the robot's ability to complete the first object rearrangement, Stage 2 focuses on training the robot to release the object and step back smoothly, as illustrated in Figure 3. This transition is critical to ensure the robot enters a standby state in preparation for manipulating the next object. To achieve this, we fine-tune the pretrained model from Stage 1 by replacing its original reward functions with a new set of objectives designed specifically for release and retreat behaviors.

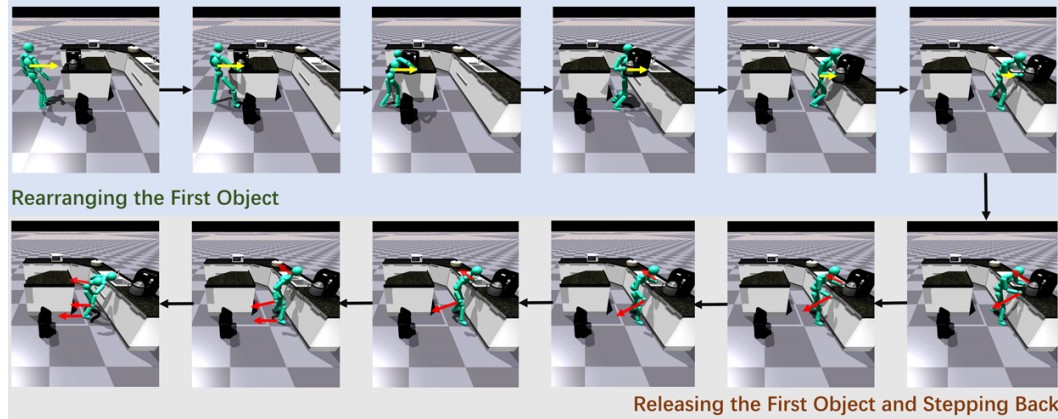

Figure 3: Two-sub-stage first object rearrangement in **HumanoidVerse**. In Sub-Stage 1 (top row), the robot places the first object at the goal location (yellow arrows). In Sub-Stage 2 (bottom row), it releases the object and steps back to a neutral position (red arrows), enabling stable transitions for the next manipulation.

The final reward function for Stage 2 is presented in Algorithm 2. The algorithm first computes the minimum distance between the robot's two hands and the object (line 1), then calculates two exponential reward terms based on the robot-to-object and hand-to-object distances (line 2). Upper-bound thresholds are applied to cap these rewards, encouraging the robot to maintain a torso-to-object distance of about 1 meter and a hand-to-object distance of about 0.5 meters (lines 3–8). These specific values are chosen because they strike a balance: the robot must learn to step back

sufficiently to avoid interfering with the object, but not so far that it risks falling over or colliding with other unrelated objects. Thus, when the distances exceed these thresholds, the rewards are set to 1 to indicate that no further incentive is needed for moving farther away. Additionally, to prevent premature or degenerate behaviors, the rewards are masked to zero if the object has not yet reached sufficiently close to its goal position (lines 9–11). The final reward is a weighted sum of the two components, encouraging the robot to release and step back only after successful object placement (line 12).

---

**Algorithm 2** Stage 2 Reward Function

---

1: $d_{\text{hand2object}} \leftarrow \min(d_{\text{hand2object}}^{\text{left}}, d_{\text{hand2object}}^{\text{right}})$
2: $r_{\text{robot2object}} \leftarrow 1 - \exp(-0.5 \cdot d_{\text{robot2object}}); \quad r_{\text{hand2object}} \leftarrow 1 - \exp(-0.5 \cdot d_{\text{hand2object}})$
3: **if** $d_{\text{robot2object}} > 1.0$ **then**
4:      $r_{\text{robot2object}} \leftarrow 1$
5: **end if**
6: **if** $d_{\text{hand2object}} > 0.5$ **then**
7:      $r_{\text{hand2object}} \leftarrow 1$
8: **end if**
9: **if** $d_{\text{object2goal}} \geq \text{threshobject2goal}$ **then**
10:      $r\text{robot2object} \leftarrow 0; , r_{\text{hand2object}} \leftarrow 0$
11: **end if**
12: $\text{reward} \leftarrow 0.5 \cdot r_{\text{robot2object}} + 0.5 \cdot r_{\text{hand2object}}$

---

### 3.3 STAGE 3: TRAINING THE SECOND TEACHER MODEL FOR REARRANGING THE SECOND OBJECT

Although the model pretrained in Stage 1 acquires the ability to approach and rearrange objects from various starting positions, it is always initialized from a standard upright posture. However, after completing the first rearrangement task, the robot often ends up in non-ideal configurations—such as bending over or even facing away from the second object. Therefore, additional training is required.

In this stage, each episode begins by using the Stage 2-trained Model 1 to complete the first object's rearrangement, including releasing the object and stepping back. Subsequently, a new Model 2 (pretrained in Stage 1) takes over and begins the rearrangement of the second object. During this phase, the loss and reward functions are applied exclusively to Model 2 and are identical to those used during its initial pretraining in Stage 1.

As illustrated in Figure 4, this training process equips Model 2 with a set of advanced capabilities. The model is able to rise from a non-standard initial posture and move toward the designated second object in a stable and coordinated manner. During this process, it demonstrates the ability to perceive and respond to the surrounding environment, allowing it to circumvent potential obstacles naturally rather than relying on an explicitly designed navigation module. Furthermore, the model can reorient itself and approach the object even when it is initially facing away, showing that it can perform the task from a backward-facing position with smooth transitions and reliable control.

By enabling the robot to initiate the second-object manipulation from irregular states and successfully execute such complex behaviors, this stage significantly improves the overall task success rate.

### 3.4 STAGE 4: DISTILLATION OF TWO TEACHER MODELS INTO A UNIFIED VLA MODEL

In Stage 4, we distill the two teacher models obtained from Stage 2 and Stage 3 into a single Vision-Language-Action (VLA) model. The original teacher models rely on oracle states—including accurate point clouds and ground-truth object poses—which are inaccessible in real-world settings. Therefore, it is necessary to distill these policies into a VLA model that only takes natural language instructions and egocentric RGB images captured by the robot's onboard camera as input and directly outputs robot actions.

We adopt the DAgger (Ross et al. (2011)) algorithm for this distillation process. A rule-based mechanism is designed to simulate a human user who monitors the robot's performance and provides the next instruction after the first task is completed. Specifically, the system determines whether the first object rearrangement task has been completed by checking three conditions: (1) the object has

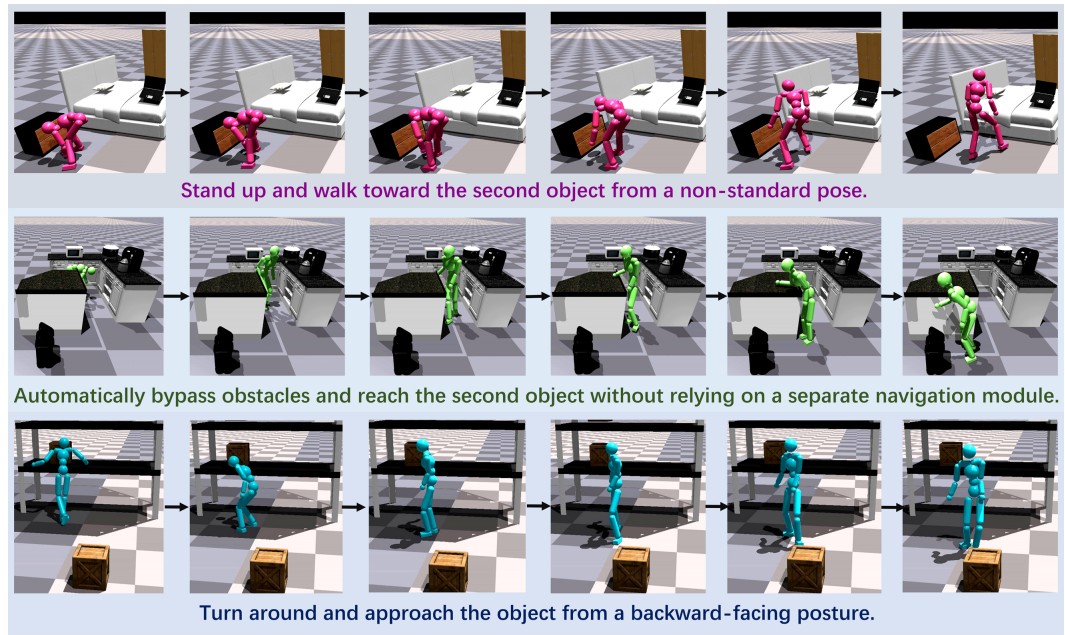

Figure 4: Second-object manipulation after Stage 3 training. Top: robot **stands up from non-standard postures** and approaches the target. Middle: it **bypasses obstacles smoothly** without explicit planning. Bottom: starting backward, it **turns around and continues toward the object**, showing robustness and adaptability.

nearly stopped moving, (2) it is sufficiently close to the goal position, and (3) the robot has stepped away from the object. This logic is summarized in Algorithm 3.

At each timestep, the system determines whether the robot is still executing the first task or has transitioned to the second. If it is still in the first task phase, the action is queried from Teacher Model 1 as the supervision target. Once the transition condition is satisfied and the second instruction is triggered, Teacher Model 2 is used instead. This dynamic teacher selection enables the VLA model to receive stage-appropriate guidance throughout the entire training process.

Upon completing this stage, the unified VLA model is capable of completing two consecutive object rearrangement tasks in a single episode while interacting with human users through natural language instructions alone.

---

**Algorithm 3** Human Instruction Simulation and Teacher Switching Logic

---

1: **Input:** object speed $s$, object-to-goal distance $d$, robot-to-object distance $r$, progress step $p$, current instruction ID $m$
2: **Output:** updated instruction ID $m'$, selected teacher model
3: **if** $s <$ speed_thresh **and** $d <$ eval_success_thresh **and** $r >$ distance_thresh **and** $p >$ time_thresh **then**
4:     $m' \leftarrow 2$                                           ▷ Trigger second instruction
5: **else**
6:     $m' \leftarrow 1$                                           ▷ Remain at first instruction
7: **end if**
8: **if** $m' = 1$ **then**
9:     Use Teacher Model 1 for supervision
10: **else**
11:     Use Teacher Model 2 for supervision
12: **end if**
13: **return** $m'$, selected teacher model

---

# 4 EXPERIMENTS

## 4.1 DATASET AND EVALUATION METRICS

We construct a dataset with 350 tasks spanning four room layouts—bedroom, kitchen, living room, and warehouse—where each task involves sequentially rearranging two different objects. Each task is accompanied by two human-written natural language instructions describing the rearrangement steps. To evaluate generalization, we curate 66 unseen tasks featuring distinct layouts, objects, and instructions, serving as a challenging out-of-distribution benchmark.

Model performance is assessed using Success 1 and Success 2—the success rates of rearranging the first and second objects respectively—and Success All, the rate of completing both. Additionally, Dist 1 and Dist 2 measure the final placement accuracy for each object. Higher success rates and lower distances indicate better performance.

## 4.2 QUANTITATIVE RESULTS

We evaluate our method on 350 multi-object rearrangement tasks, with results summarized in Table 1.

In the upper half, our **HumanoidVerse-Teacher** is compared with **HumanVLA-Teacher** (Xu et al., 2024). On the first-object task (*Success 1*), our model achieves 88.762% success—over twice HumanVLA's 42.286%—benefiting from Stage 1 and Stage 2 training. For the second-object task (*Success 2*), our method reaches 72.857%, far surpassing HumanVLA's 7.429%, owing to Stage 2's object-release and step-back training and Stage 3's robustness to varied initial configurations. The rate for completing both tasks (*Success All*) is 72.381%, nearly matching *Success 2*, showing minimal disturbance to the first object—unlike HumanVLA's 5.714%. In placement accuracy, our method achieves lower errors: 0.247 m vs. 0.468 m for *Dist 1*, and 0.484 m vs. 1.793 m for *Dist 2*.

The lower half reports **student** results. Despite the difficulty of sequential two-object tasks for policy distillation, our Stage 4 *dual-teacher state-transition distillation* yields substantial gains: *Success 1* 66.095% (vs. 45.714%), *Success 2* 42.667% (vs. 0.000%), and *Success All* 40.952% (vs. 0.000%). Placement errors are 0.503 m (vs. 0.996 m) for the first object and 1.031 m (vs. 1.971 m) for the second. These results demonstrate that our method enables effective distillation in a challenging multi-stage setting, consistently outperforming baselines in both success rate and spatial precision.

Table 1: Performance comparison between HumanVLA and our HumanoidVerse models on 350 multi-object rearrangement tasks.

|  | Success 1 ↑ | Success 2 ↑ | Success All ↑ | Dist 1 (m) ↓ | Dist 2 (m) ↓ |
|---|---|---|---|---|---|
| HumanVLA-T | 42.286% | 7.429% | 5.714% | 0.468 | 1.793 |
| **HumanoidVerse-T** | **88.762%** | **72.857%** | **72.381%** | **0.247** | **0.484** |
| HumanVLA-S | 45.714% | 0.000% | 0.000% | 0.996 | 1.971 |
| **HumanoidVerse-S** | **66.095%** | **42.667%** | **40.952%** | **0.503** | **1.031** |

In addition to the above comparisons, we further conduct experiments on the teacher model to examine whether alternative training paradigms could match the performance of our staged HumanoidVerse-Teacher.

First, we build an end-to-end RL training framework that directly trains a single policy to complete the full two-object rearrangement task without staged decomposition. Despite extensive tuning, the end-to-end policy struggles to make meaningful progress on both-object manipulation and exhibits very poor spatial precision.

Second, we evaluate a single-teacher baseline: after completing the training for the first object ("Teacher One"), we continue RL training on the second-object task using the same teacher model. This baseline reflects whether a strong single-task teacher can be naturally extended to multi-object sequential manipulation.

The quantitative results are summarized in Table 2. The end-to-end RL approach fails almost completely on multi-step manipulation, achieving only 2.286% on Success 1 and 0% on Success 2, with placement errors exceeding 1.96 m. The single-teacher baseline performs better, demonstrating that continuing training on the second task can yield moderate performance (e.g., 48.476% Success 2). However, it still falls far short of our multi-stage HumanoidVerse-Teacher, especially on second-object success and spatial accuracy. These comparisons further validate the necessity of our staged and dual-teacher design and show that HumanoidVerse substantially outperforms both naive end-to-end RL and simple single-teacher fine-tuning.

Table 2: Teacher-model comparisons on 350 multi-object rearrangement tasks.

| | Success 1 ↑ | Success 2 ↑ | Success All ↑ | Dist 1 (m) ↓ | Dist 2 (m) ↓ |
|---|---|---|---|---|---|
| End-to-End RL | 2.286% | 0.000% | 0.000% | 2.050 | 1.966 |
| Single Teacher | 88.286% | 48.476% | 48.190% | 0.258 | 1.014 |
| **HumanoidVerse-T** | **88.762%** | **72.857%** | **72.381%** | **0.247** | **0.484** |

## 4.3 QUANTITATIVE RESULTS ON UNSEEN TASKS

Table 3 summarizes the performance of our method, **HumanoidVerse**, on 66 previously unseen multi-object rearrangement tasks that vary in environment, objects, and instructions.

Compared to HumanVLA (Xu et al., 2024), HumanoidVerse-Teacher delivers substantially better results. It achieves 80.303%, 64.646%, and 64.646% for Success 1, Success 2, and Success All, respectively, while HumanVLA-Teacher lags behind with 27.273%, 9.091%, and 4.545%. In addition, our model places objects more precisely, with average distance errors of 0.531 m and 0.723 m, compared to HumanVLA's 0.790 m and 1.848 m.

For student models, HumanoidVerse-S sustains high effectiveness, scoring 56.061%, 42.929%, and 40.909% on the three success metrics. Remarkably, its overall success barely declines from the training tasks, highlighting strong adaptability. By contrast, HumanVLA-S fails to handle multi-step sequences in these new scenarios, achieving zero success on Success 2 and Success All, and exhibits notably larger placement errors (1.271 m and 2.043 m) than our approach (0.771 m and 1.226 m).

Overall, these findings emphasize the enhanced robustness and precision of HumanoidVerse on challenging unseen tasks.

Table 3: Results on 66 **unseen** multi-object rearrangement tasks. HumanoidVerse surpasses HumanVLA across success and placement metrics for both teacher and student models.

| | Success 1 ↑ | Success 2 ↑ | Success All ↑ | Dist 1 (m) ↓ | Dist 2 (m) ↓ |
|---|---|---|---|---|---|
| HumanVLA-T | 27.273% | 9.091% | 4.545% | 0.790 | 1.848 |
| **HumanoidVerse-T** | **80.303%** | **64.646%** | **64.646%** | **0.531** | **0.723** |
| HumanVLA-S | 37.879% | 0.000% | 0.000% | 1.271 | 2.043 |
| **HumanoidVerse-S** | **56.061%** | **42.929%** | **40.909%** | **0.771** | **1.226** |

## 4.4 ABLATION STUDY

Table 4 reports ablation results on 350 two-object rearrangement tasks using the HumanoidVerse-Teacher model.

Removing **Training Stage 2** (step-back and object release) reduces *Success 1* from 88.762% to 81.048%, and causes sharper drops in *Success 2* and *Success All* to 60.476% and 56.190%, indicating its key role in second-object manipulation.

Without **Training Stage 3** (diverse start states), *Success 1* slightly decreases to 86.952%, but *Success 2* and *Success All* fall dramatically to 35%, showing Stage 3's critical importance.

Omitting the object speed condition in the dual-teacher switching lowers *Success 1* to 68.952% and *Success All* to 53.524%, suggesting premature transitions before object stabilization.

Removing the object-to-goal distance condition leads to *Success 1* dropping to 70.286%, with *Success 2* increasing to 77.238%, but overall success declines to 55.048%, implying unstable placements affect final outcomes.

Table 4: Ablation study on 350 two-object tasks, showing the impact of removing training stages or switching conditions from HumanoidVerse-Teacher.

| | Success 1 ↑ | Success 2 ↑ | Success All ↑ |
|---|---|---|---|
| **HumanoidVerse-T** | **88.762%** | 72.857% | **72.381%** |
| w/o Training S2 | 81.048% | 60.476% | 56.190% |
| w/o Training S3 | 86.952% | 35.238% | 34.952% |
| w/o Obj Speed | 68.952% | 69.238% | 53.524% |
| w/o Obj2Goal Dist | 70.286% | **77.238%** | 55.048% |

To further examine whether the policy relies disproportionately on any single input modality, we conduct an additional ablation study that selectively removes one of the three modalities—vision, language, or proprioception—while keeping all other components unchanged. This test evaluates whether HumanoidVerse-Teacher truly requires multimodal information for robust manipulation, or if the policy could compensate when one input stream is absent.

As shown in Table 5, removing any individual modality results in near-complete task failure: both Success 1 and Success 2 drop to almost zero, and placement errors exceed 2 m. These results demonstrate that all three modalities contribute critically and complementarily to the policy's decision-making process. In particular, the absence of vision or proprioception prevents meaningful object interaction, while removing language eliminates goal grounding. Together, these findings confirm that multimodal fusion is essential for achieving reliable performance in HumanoidVerse.

Table 5: Ablation on input modalities for HumanoidVerse-Student, showing the effect of removing vision, language, or proprioception.

| | Success 1 ↑ | Success 2 ↑ | Success All ↑ | Dist 1 (m) ↓ | Dist 2 (m) ↓ |
|---|---|---|---|---|---|
| **HumanoidVerse-S** | **66.095%** | **42.667%** | **40.952%** | **0.503** | **1.031** |
| w/o Vision | 0.000% | 0.000% | 0.000% | 2.143 | 1.988 |
| w/o Language | 0.287% | 0.000% | 0.000% | 2.298 | 2.132 |
| w/o proprioception | 0.000% | 0.000% | 0.000% | 2.148 | 1.964 |

## 5 CONCLUSION

We present HumanoidVerse, the first vision-language guided humanoid framework capable of performing long-horizon, multi-object rearrangement tasks in diverse and realistic environments. Our system integrates a multi-stage teacher-student training pipeline with dynamic teacher switching, enabling seamless transitions between tasks without destabilizing the environment. Through extensive experiments, HumanoidVerse demonstrates strong performance and generalization across 350 training and 66 unseen tasks, significantly outperforming existing baselines in both success rates and spatial accuracy. By bridging high-level natural language commands and low-level physical control, our work advances the frontier of general-purpose humanoid agents and opens new possibilities for interactive, instruction-driven robot autonomy in complex real-world settings.

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

# A  APPENDIX

## A.1  REWARD FUNCTION

To train the agent to successfully complete the rearrangement task, we formulate the task reward in *Training Stage 1* and *Training Stage 3* as a weighted sum of shaped sub-rewards that guide the agent through the phases of approaching, grasping, and transporting an object.

The overall task reward is expressed as:

$$r_{\text{task}} = \sum_{i=1}^{N} \alpha_i \cdot r_i \tag{3}$$

where $\alpha_i$ is the weight and $r_i$ is the $i$-th reward component.

The reward components used in our system are listed in Table 6, with each component defined formally in Table 7.

Table 6: Descriptions and weights of active reward terms

| Weight $\alpha_i$ | Reward Name | Description |
|---|---|---|
| 0.1 | $r_{\text{robot2object\_vel}}$ | Robot's velocity aligned toward object or guide |
| 0.1 | $r_{\text{robot2object\_pos}}$ | Distance between robot and object (or guide) |
| 0.1 | $r_{\text{hand2object}}$ | Minimum hand-to-object point cloud distance |
| 0.1 | $r_{\text{height}}$ | Object lifted to appropriate carry height |
| 0.2 | $r_{\text{object2goal\_vel}}$ | Object velocity directed toward final goal |
| 0.2 | $r_{\text{object2goal\_pos\_far}}$ | Distance from object to intermediate guide point |
| 0.1 | $r_{\text{object2goal\_pos\_near}}$ | Distance from object to final goal |
| 0.1 | $r_{\text{object2goal\_rot}}$ | Orientation alignment between object and goal |

Table 7: Mathematical definitions of reward components

| Reward Name | Formula |
|---|---|
| $r_{\text{robot2object\_vel}}$ | $\exp\left(-2 \cdot (v_t - \mathbf{v}_r \cdot \hat{\mathbf{d}})^2\right)$ |
| $r_{\text{robot2object\_pos}}$ | $\exp\left(-0.5 \cdot \|\mathbf{p}_r - \mathbf{p}_o\|^2\right)$ |
| $r_{\text{hand2object}}$ | $\mathbb{E}_j\left[\exp\left(-5 \cdot \min_j \|\mathbf{p}_h - \mathbf{p}_j^{\text{pcd}}\|\right)\right]$ |
| $r_{\text{height}}$ | $\frac{\min(z_o, z_{\text{target}}) - z_{\text{init}}}{z_{\text{target}} - z_{\text{init}}}$ |
| $r_{\text{object2goal\_vel}}$ | $\exp\left(-2 \cdot (v_o^{\text{target}} - \mathbf{v}_o \cdot \hat{\mathbf{d}})^2\right)$ |
| $r_{\text{object2goal\_pos\_far}}$ | $\exp\left(-1 \cdot \|\mathbf{p}_o - \mathbf{p}_{g,\text{inter}}\|\right)$ |
| $r_{\text{object2goal\_pos\_near}}$ | $\exp\left(-5 \cdot \|\mathbf{p}_o - \mathbf{p}_g\|\right)$ |
| $r_{\text{object2goal\_rot}}$ | $\exp\left(-2 \cdot \Delta R(\mathbf{q}_o, \mathbf{q}_g)\right)$ |

**Notation:**

- $\mathbf{p}_r, \mathbf{p}_o, \mathbf{p}_g$: positions of the robot, object, and goal.
- $\mathbf{v}_r, \mathbf{v}_o$: velocities of the robot and object.
- $\hat{\mathbf{d}}$: unit vector toward current target (object or guide).

- $v_t$, $v_o^{\text{target}}$: target speeds (typically 1.5).
- $\mathbf{p}_h$: robot hand position.
- $\mathbf{p}_j^{\text{pcd}}$: point cloud point on the object.
- $z_o$, $z_{\text{init}}$, $z_{\text{target}}$: object's current, initial, and desired vertical positions.
- $\Delta R(\mathbf{q}_o, \mathbf{q}_g)$: rotational difference between object and goal (e.g., geodesic quaternion distance).

## A.2 TRAINING EPOCHS FOR EACH STAGE

The number of training epochs allocated to each stage is shown in Table 8. These values were chosen based on observed learning plateaus and performance stabilization criteria.

Table 8: Number of training epochs per curriculum stage

| Stage | Epochs | Description |
|---|---|---|
| 1 | 28,200 | Pretraining for Single-Object Rearrangement |
| 2 | 550 | Learning to Release and Step Back After the First Rearrangement |
| 3 | 18,550 | Training the second teacher model for rearranging the second object |
| 4 | 104,000 | Distillation of Two Teacher Models into a Unified VLA Model |

## A.3 OBSERVATION AND ACTION SPACE

We adopt the default observation and action space design of HumanVLA Xu et al. (2024), covering proprioception, object state, goal specification, a waypoint signal, and projected discriminator observations. Tables 9-13 summarize all components.

Table 9: Proprioceptive observation space (223 dimensions).

| Feature | Dim. |
|---|---|
| Root height | $\mathbb{R}^1$ |
| Root linear velocity | $\mathbb{R}^3$ |
| Root angular velocity | $\mathbb{R}^3$ |
| Root rotation (6D representation) | $\mathbb{R}^6$ |
| Link positions (14 links) | $\mathbb{R}^{14\times3}$ |
| Link rotations (14 links, 6D) | $\mathbb{R}^{14\times6}$ |
| Link linear velocities | $\mathbb{R}^{14\times3}$ |
| Link angular velocities | $\mathbb{R}^{14\times3}$ |

**Adopted Setting.** We directly follow the default observation and action configuration of HumanVLA when constructing our HumanoidVerse system.

## A.4 QUALITATIVE COMPARISON BETWEEN HUMANOIDVERSE AND HUMANVLA

Figure 5 illustrates the performance of HumanoidVerse in executing two consecutive natural language instructions: "Move the coffeemaker from the kitchen island to the right side of the countertop, near the sink." and "Move the trashbin from the right side of the countertop to the left side of the countertop." Each sub-image in the figure shows a split view: the left half displays a third-person

Table 10: Object observation space.

| Feature | Dim. |
|---|---|
| Object position | $\mathbb{R}^3$ |
| Object rotation (6D representation) | $\mathbb{R}^6$ |
| Object linear velocity | $\mathbb{R}^3$ |
| Object angular velocity | $\mathbb{R}^3$ |
| BPS geometry (200 basis points) | $\mathbb{R}^{200 \times 3}$ |

Table 11: Goal and waypoint representation.

| Feature | Dim. |
|---|---|
| Goal position | $\mathbb{R}^3$ |
| Goal rotation (6D representation) | $\mathbb{R}^6$ |
| Waypoint $x_t^{wp}$ | $\mathbb{R}^3$ |

Table 12: Projected discriminator observation (per frame), following AMP Peng et al. (2021). Ten frames are stacked to obtain $\mathbb{R}^{10 \times 131}$.

| Feature | Dim. |
|---|---|
| Root height | $\mathbb{R}^1$ |
| Root linear velocity | $\mathbb{R}^3$ |
| Root angular velocity | $\mathbb{R}^3$ |
| Root rotation (6D representation) | $\mathbb{R}^6$ |
| Joint rotations (12 joints, 6D) | $\mathbb{R}^{12 \times 6}$ |
| Joint velocities | $\mathbb{R}^{28 \times 1}$ |
| End-effector positions (hands, feet, head) | $\mathbb{R}^{5 \times 3}$ |
| Object position | $\mathbb{R}^3$ |

Table 13: Action space.

| Action Type | Dim. |
|---|---|
| PD controller target joint positions | $\mathbb{R}^{28}$ |

perspective capturing the robot's full-body motion, while the right half presents the egocentric view from the robot's onboard camera, as its visual perception.

As shown, HumanoidVerse (Ours) successfully completes the first instruction and patiently waits for the second. Upon receiving the second instruction, the robot smoothly turns and carries it out without issue. In contrast, Figure 6 presents the performance of HumanVLA under the same task. It can be observed that the robot struggles to handle the sequential instructions: during the execution, it collides with both the trashbin and the coffeemaker, leading to an unfamiliar and unstable state. As a result, it fails to complete even the first instruction, let alone the second. This comparison high-

lights the superior robustness and compositional reasoning capability of HumanoidVerse in handling complex, multi-step embodied tasks.

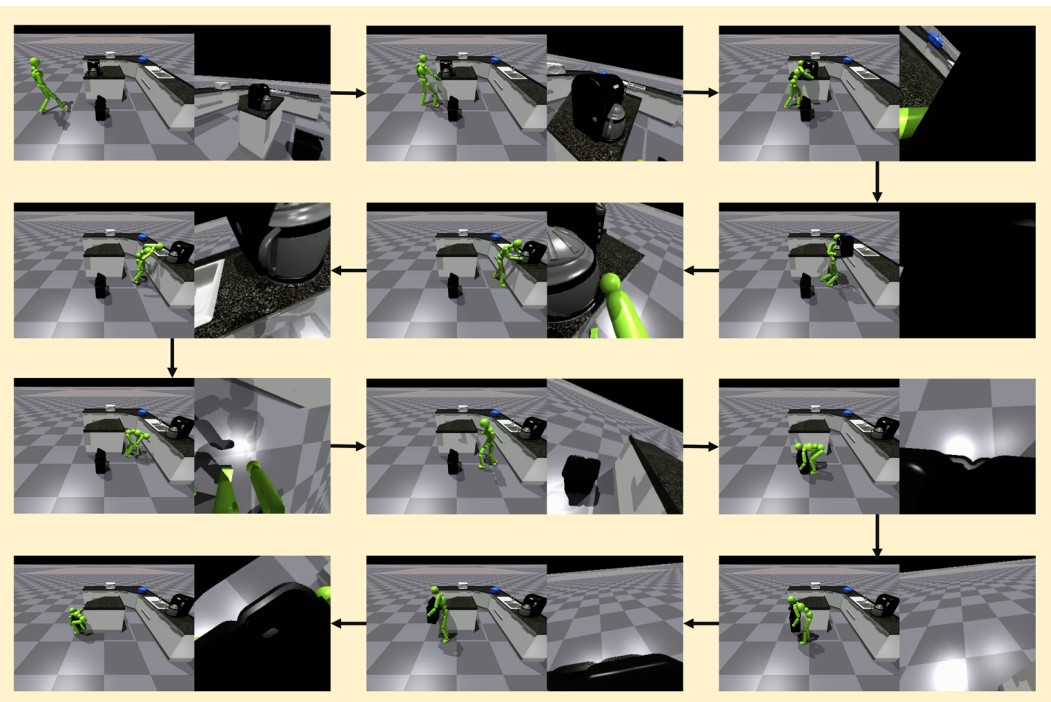

Figure 5: Execution visualization of HumanoidVerse (Ours) on two consecutive instructions: (1) "Move the coffeemaker from the kitchen island to the right side of the countertop, near the sink," and (2) "Move the trashbin from the right side of the countertop to the left side of the countertop." Each sub-image pair displays a third-person view (left) and the egocentric camera view of the robot (right). HumanoidVerse successfully completes both instructions with smooth transitions and accurate object manipulation, demonstrating robust understanding and compositional reasoning.

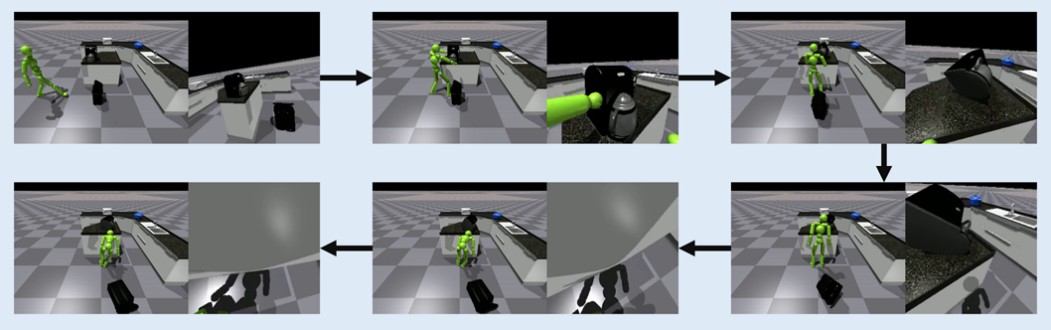

Figure 6: Execution visualization of HumanVLA on the same task as in Figure 6. The robot receives the same two consecutive instructions. However, during execution, it collides with both the coffeemaker and the trashbin, causing object displacement and entering an unfamiliar state. This leads to task failure—HumanVLA is unable to complete even the first instruction. Each sub-image pair shows the third-person view (left) and the egocentric view (right). The failure highlights HumanVLA's limitations in handling sequential object manipulation tasks under realistic embodied conditions.

### A.5 MORE QUALITATIVE RESULTS

This section presents additional visualizations of execution examples from HumanoidVerse in a variety of environments, including bedroom, kitchen, living room, and warehouse. Figure 7, 8, 9, 10, 11, 12 and 13 illustrate these examples.

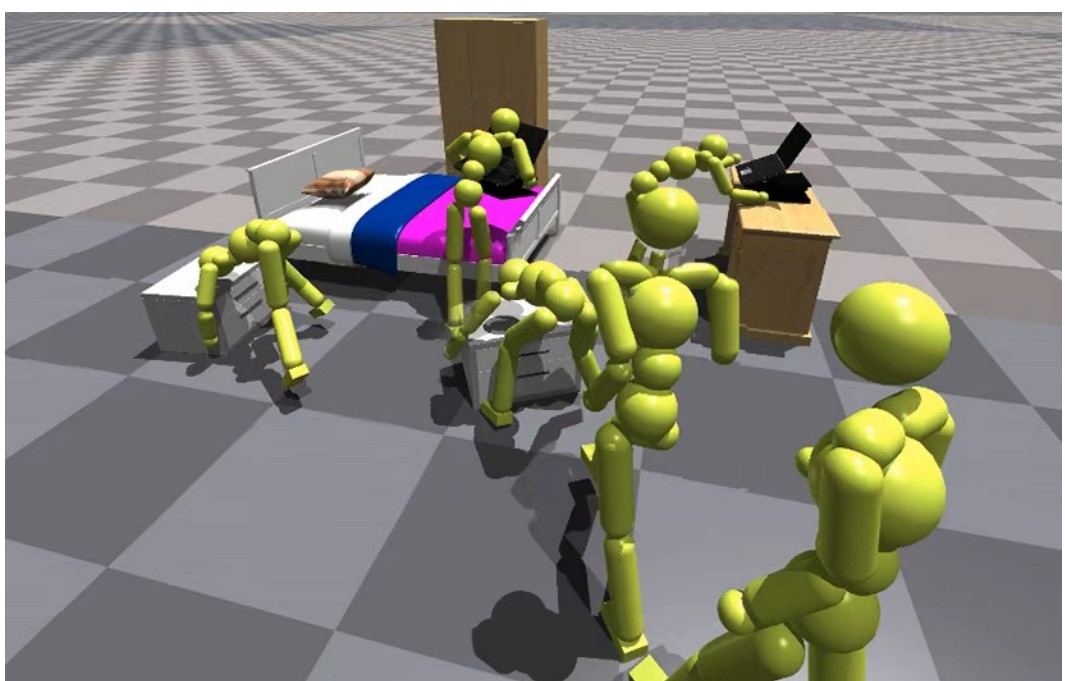

Figure 7: Visualization of humanoid execution in the bedroom scenario, successfully completing rearrangement of two objects (the bedside table and the laptop).

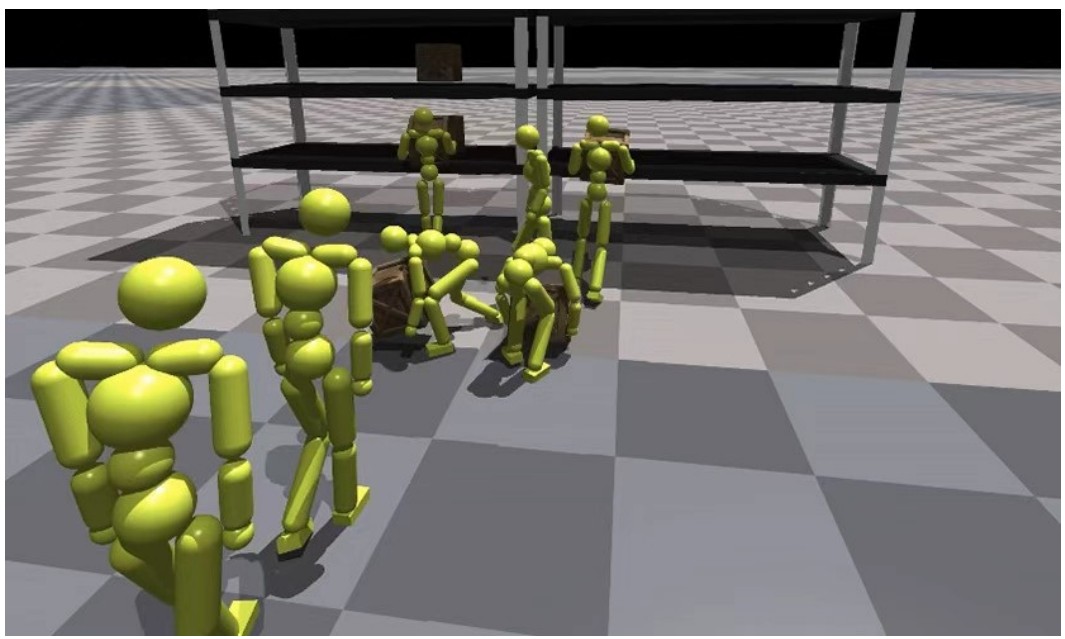

Figure 8: Visualization of humanoid execution in the warehouse scenario, successfully completing rearrangement of two boxes.

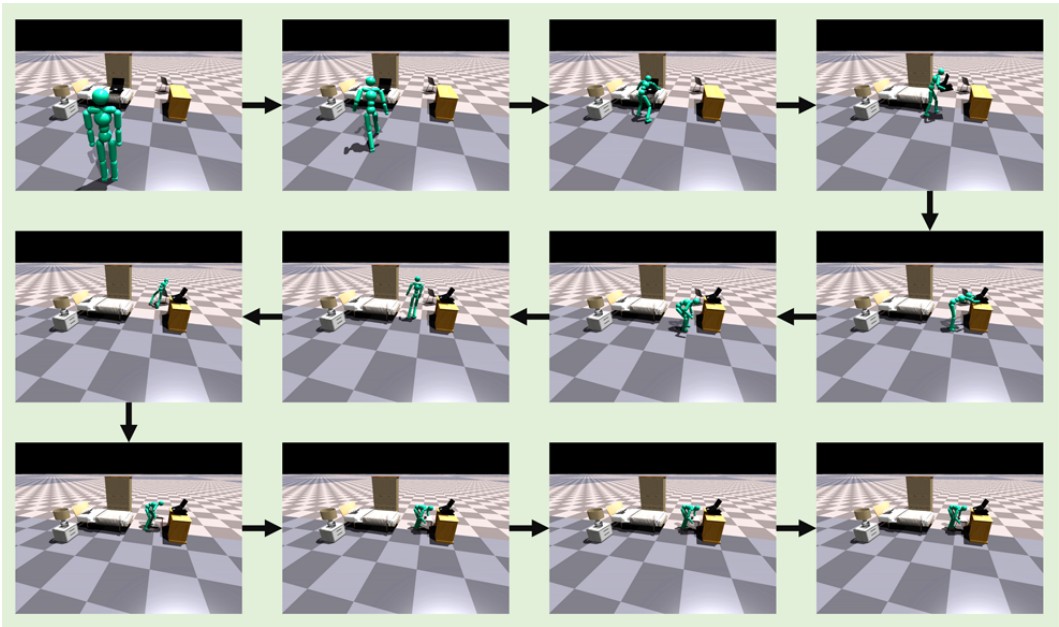

Figure 9: Visualization of humanoid execution in the bedroom scenario based on two sequential text instructions. The first instruction is: "Move the laptop from the bed to the desk on the right side of the room." The second instruction is: "Move the chair from the left side of the desk to the front side of the desk." The figure shows the humanoid successfully completing both tasks through a sequence of interactions with objects in the environment.

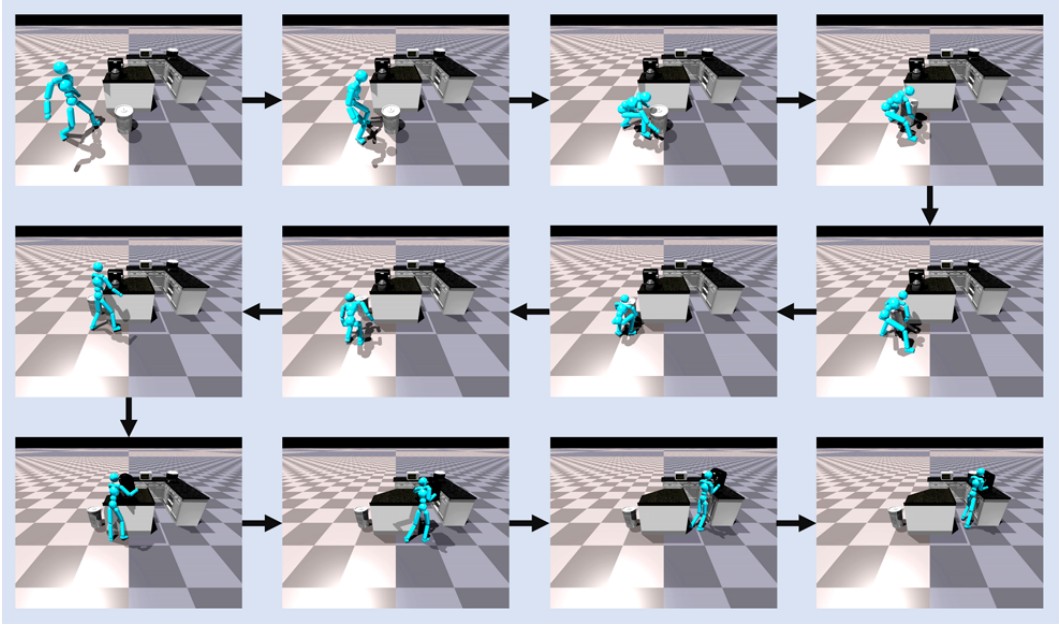

Figure 10: Visualization of humanoid execution in the kitchen scenario based on two sequential text instructions. The first instruction is: "Move the trashbin from the right side of the kitchen island to the left side of the kitchen island." The second instruction is: "Move the coffeemaker from the right side of the kitchen island to the right side of the countertop."

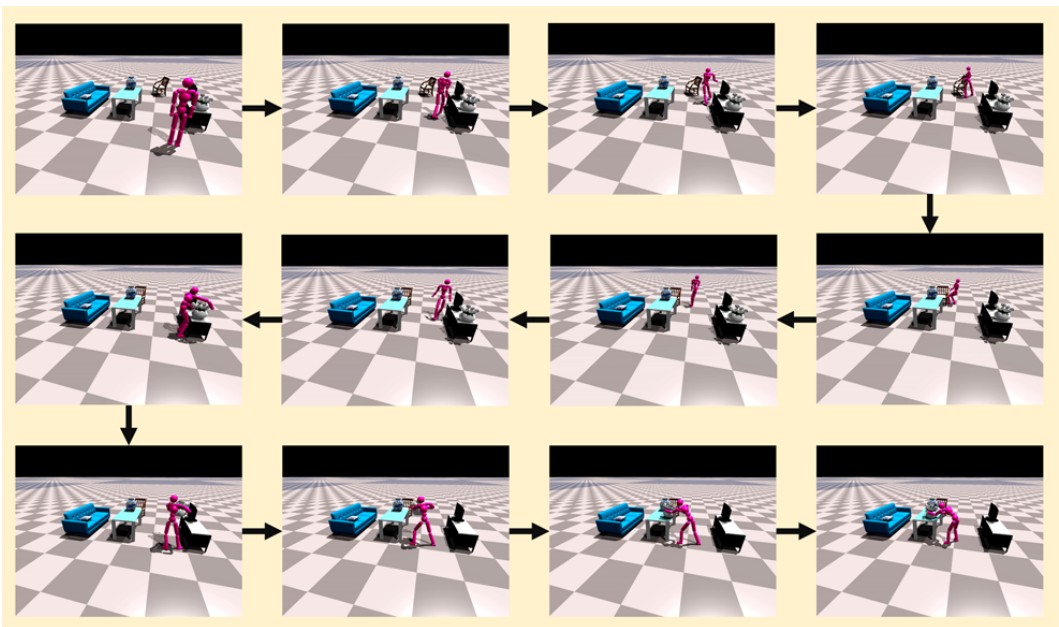

Figure 11: Visualization of humanoid execution in the kitchen scenario based on two sequential text instructions. The first instruction is: "Move the chair to the right side of the console table." The second instruction is: "Move the plant from the left side of the tv cabinet to the center of the coffee table."

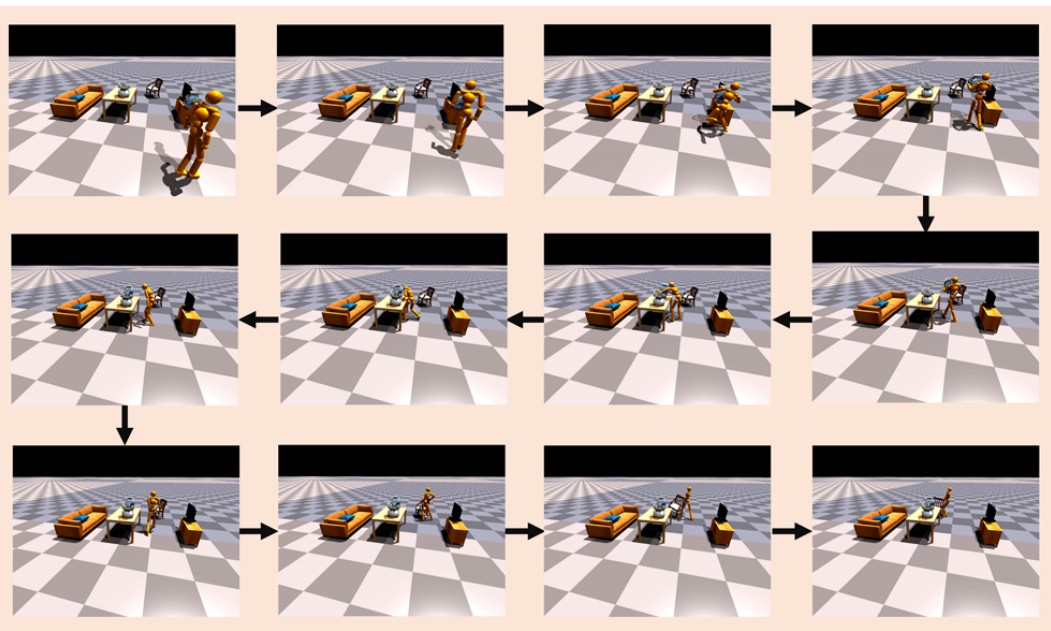

Figure 12: Visualization of humanoid execution in the kitchen scenario based on two sequential text instructions. The first instruction is: "Move the vase from the tv stand to the coffee table." The second instruction is: "Move the chair from its position to the right side of the table."

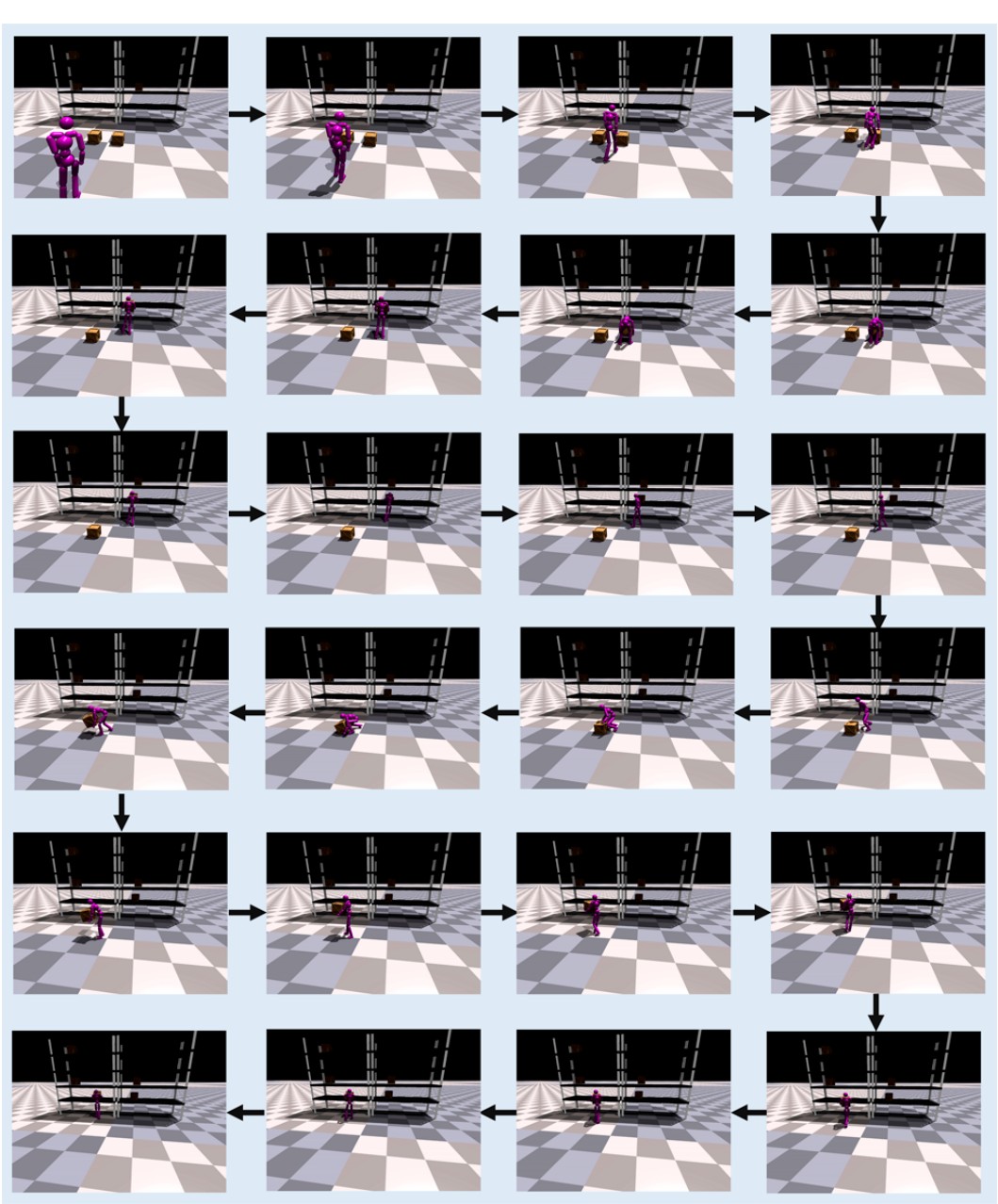

Figure 13: Visualization of humanoid execution in the kitchen scenario based on two sequential text instructions. The first instruction is: "Move the right box from the ground to the left of the right shelf." The second instruction is: "Lift the other box from the ground to the right of the left shelf."

## B  THE USE OF LARGE LANGUAGE MODELS (LLMs)

In the preparation of this manuscript, large language models (LLMs) were employed exclusively for the purposes of grammar correction, spelling verification, and enhancing sentence fluency. The models were not involved in generating ideas, analyzing data, designing experiments, or drawing conclusions. All intellectual contributions, including the formulation of research questions, methodology, results, and interpretations, are entirely attributable to the authors. The role of LLMs was strictly limited to linguistic refinement in order to improve readability and clarity of expression.

