# OpenReview forum: "HumanoidVerse: A Versatile Humanoid for Vision-Language Guided Multi-Object Rearrangement"
_ICLR.cc/2026/Conference — Submitted to ICLR 2026_

### Official Review · Reviewer_MbxK · 2025-10-31

**Soundness:** 4
**Presentation:** 3
**Contribution:** 3
**Rating:** 4
**Confidence:** 4

**Summary:**

The paper introduces HumanoidVerse, a framework for a simulated humanoid robot to perform "long-horizon, multi-object rearrangement tasks." The core contribution is a 4-stage curriculum that distills two separate teacher policies into a single VLA model: one for distill during the initial rearrangement + releasing and the other for the second object rearrangement. First, two expert "teacher" policies are trained using reinforcement learning with privileged state information (e.g., object poses). The key idea is that the second teacher (Stage 3) is specifically trained to handle the diverse, non-standard starting poses left after the first teacher completes its task, whereas the first teacher (Stage 2) is trained to let go and step back after the first object rearrangement. Both teachers are then distilled using DAgger into a single student Vision-Language-Action (VLA) model that operates from only egocentric RGB video and natural language instructions. The authors demonstrate how their framework can effectively tackle sequential humanoid object rearrangement tasks on Isaac Gym.

**Strengths:**

- **Novel Problem Formulation**: The paper's main strength is tackling continuous, sequential, multi-object manipulation, which is often overlooked in existing works on VLA models for humanoid loco-manipulation. This is a clear and important step beyond the single-object, fixed-start tasks prevalent in prior work (e.g., the HumanVLA baseline).

- **Effective Curriculum**: The 4-stage pipeline is a well-engineered solution. The ablation study (Table 3) clearly validates the authors' design, showing that explicitly training for the transition (Stage 2 for stepping back) and the second task's varied starts (Stage 3 for handling diverse initial configurations) are critical for effective training.

- **New Benchmark for Humanoid Sequential Object Rearrangement**: The creation of a benchmark dataset with 350 sequential two-object tasks is a useful contribution for future research in this area.

**Weaknesses:**

- **Limited Algorithmic Novelty and Scalability**: The paper's contribution is a highly specific training curriculum, not a new or generalizable algorithm. This curriculum is only demonstrated for a two-step ($N=2$) task, and all task involves sequentially rearranging two different objects. The paper does not provide a clear path for scaling this "dual-teacher" framework to $N>2$ tasks. This seems to imply a non-scalable $N$-teacher pipeline, which contradicts the "long-horizon" claim.

- **Weak Baseline Comparison**: The primary baseline is HumanVLA, a model designed for single-object tasks. As shown in Table 2, it achieves a 0.000% success rate on the sequential task. This is expected and only confirms the new task is harder; it does not validate that the proposed dual-teacher curriculum is a better method for sequential tasks than other plausible baselines (e.g., end-to-end RL and a hierarchical RL, or subsequently their distilled policies).

- **Simulation-Only**: All results are in simulation. The paper does not discuss the significant sim-to-real gap for a complex policy that must simultaneously handle locomotion, manipulation, and perception from RGB data.

- **No Failure Modes**: It would be nice to provide analyses on failures modes of both HumanVLA and HumanoidVerse to justify some design choices or mention potential rooms for improvements.

**Questions:**

- **Scalability**: How do you propose to scale this framework beyond $N=2$? Does your "dual-teacher" approach not become an "N-teacher" model, and if so, do you consider that a scalable solution for "long-horizon" tasks?

- **Baselines**: Why were more relevant baselines for sequential tasks, such as a monolithic end-to-end teacher or a standard hierarchical RL approach, not included in the comparison?

- **Failure Modes**: What are the common failure modes in simulated tasks? Were there any failures around the transitions between the first and second tasks?

---

> ### Author Response · Authors · 2025-11-19
> **Response to Reviewer MbxK (1/2)**
>
> We appreciate the reviewer’s comments and address all points individually as follows.
>
> > How do you propose to scale this framework beyond ? Does your "dual-teacher" approach not become an "N-teacher" model, and if so, do you consider that a scalable solution for "long-horizon" tasks?
>
> Thank you for raising this point. Existing works such as HumanVLA[1], SkillMimic[2], INTERMIMIC[3], TokenHSI[4], CooHOI[5], InterScene[6], and SuperPADL[7] focus on single-object interaction, and some do not model explicit object–object interaction at all. In contrast, our work takes an important step forward by training a robot to perform consecutive rearrangement involving two distinct objects. To the best of our knowledge, this capability has not been demonstrated before in the literature. Our training covers 350 room scenes and 79 objects, enabling a significantly more complex manipulation space compared to prior work.
>
> Regarding extension to scenarios with three or more objects: our current framework is developed and evaluated for two-object sequences, and we aim to establish a dedicated baseline for multi-object sequences (3+ objects) in future work.
>
> > Why were more relevant baselines for sequential tasks, such as a monolithic end-to-end teacher or a standard hierarchical RL approach, not included in the comparison?
>
> Thank you for the question. We have added experimental results comparing the end-to-end RL method and the single-teacher approach in Section 4.2 (Quantitative Results), Table 2. The results show that both the success rates and distance metrics drop significantly under the end-to-end RL and single-teacher paradigms. This demonstrates that the dual-teacher design is critical for achieving high performance, as it effectively mitigates failure cases that arise when relying on a single teacher—for example, confusing the two tasks or over-focusing on the first task and failing to progress to the second.
>
> > What are the common failure modes in simulated tasks? Were there any failures around the transitions between the first and second tasks?
>
> Thank you for the question. One observed failure occurs in the task “Move the pot.” We hypothesize that this is due to the training scenes: both the pot and the coffee maker are often placed on a short stand, close to each other, and they appear visually similar. Because the coffee maker is box-shaped, it is easier for the robot to grasp and move, so the policy tends to prioritize moving the coffee maker first. When the robot later attempts to move the pot, it tries to apply the same manipulation strategy. However, since the pot is round and cannot be handled in the same way, this can lead to occasional failures.
>
> It is important to note that such failures are relatively rare—the task only exhibits slightly more failures than other tasks, and the overall performance remains high. This suggests that the learned policy is generally robust, with failures arising primarily in edge cases where object shape and manipulation strategy mismatch.

---

> ### Author Response · Authors · 2025-11-19
> **Response to Reviewer MbxK (2/2)**
>
> > Simulation-Only
>
> Thank you for the question. Existing works such as HumanVLA [1], SkillMimic [2], INTERMIMIC [3], TokenHSI [4], CooHOI [5], InterScene [6], and SuperPADL [7] have all conducted experiments in simulation but not in reality.
>
> In contrast, our work tackles a much more diverse and complex set of scenarios—including 350 different room scenes, multi-object rearrangement tasks, and a wide variety of object types—using a single general-purpose model. To the best of our knowledge, no prior work has performed real-robot experiments at this scale. We plan to extend our research to real humanoid robots in future work, which will involve addressing hardware differences, perception noise, and environment variability.
>
> [1] Xinyu Xu, Yizheng Zhang, Yong-Lu Li, Lei Han, and Cewu Lu. Humanvla: Towards vision-language directed object rearrangement by physical humanoid. Advances in Neural Information Processing Systems, 37:18633–18659, 2024.
>
> [2] Yinhuai Wang, Qihan Zhao, Runyi Yu, Ailing Zeng, Jing Lin, Zhengyi Luo, Hok Wai Tsui, Jiwen Yu, Xiu Li, Qifeng Chen, et al. Skillmimic: Learning reusable basketball skills from demonstrations. arXiv e-prints, pp. arXiv–2408, 2024.
>
> [3] Sirui Xu, Hung Yu Ling, Yu-Xiong Wang, and Liang-Yan Gui. Intermimic: Towards universal whole-body control for physics-based human-object interactions. In Proceedings of the Computer Vision and Pattern Recognition Conference, pp. 12266–12277, 2025.
>
> [4] Liang Pan, Zeshi Yang, Zhiyang Dou, Wenjia Wang, Buzhen Huang, Bo Dai, Taku Komura, and Jingbo Wang. Tokenhsi: Unified synthesis of physical human-scene interactions through task tokenization. In Proceedings of the Computer Vision and Pattern Recognition Conference, pp.5379–5391, 2025.
>
> [5] Gao, Jiawei and Wang, Ziqin and Xiao, Zeqi and Wang, Jingbo and Wang, Tai and Cao, Jinkun and Hu, Xiaolin and Liu, Si and Dai, Jifeng and Pang, Jiangmiao. Coohoi: Learning cooperative human-object interaction with manipulated object dynamics. In Advances in Neural Information Processing Systems, pp.79741--79763, 2024.
>
> [6] Liang Pan, Jingbo Wang, Buzhen Huang, Junyu Zhang, Haofan Wang, Xu Tang, and Yangang Wang. Synthesizing physically plausible human motions in 3d scenes. In 2024 International Conference on 3D Vision (3DV), pp. 1498–1507. IEEE, 2024.
>
> [7] Juravsky, Jordan and Guo, Yunrong and Fidler, Sanja and Peng, Xue Bin. Superpadl: Scaling language-directed physics-based control with progressive supervised distillation. In ACM SIGGRAPH 2024 Conference Papers, pp. 1--11, 2024.

---

### Official Review · Reviewer_YEVG · 2025-10-31

**Soundness:** 2
**Presentation:** 3
**Contribution:** 2
**Rating:** 4
**Confidence:** 3

**Summary:**

HumanoidVerse presents a method that trains VLA models to control simulated humanoid robots to perform multi-object rearrangement. It consists of a multi-stage training process to rearrange the first object, step away from the first object, and rearrange the second object. The authors distill this into a VLA using DAgger, such that it operates using only image and language input (no privileged information) at test-time.

**Strengths:**

- The success rate on multi-object sequences increases dramatically over the HumanVLA model the authors continue training from.
- The method appropriately drops privileged conditioning, representing a plausible input-output setup for real-world deployment.
- Distilling privileged teachers into a VLA with general world knowledge is an approach that can introduce additional robustness and sim data to VLAs and is not limited to teleoperated data collection.
- The paper is clearly written and easy to follow.

**Weaknesses:**

- The biggest drawback of the paper is the limited scope of the task, which conflicts with the central claim of the method being "a key step toward robust, general-purpose humanoid agents." The reinforced rewards are very hand-designed with rule-based triggers and remedies to prevent observed, undesirable behavior. This is directionally opposite of scalable learning.
- For the above reason, the improvements are likely isolated to the multi-stage object rearrangement setting.
- There isn't discussion about inference latency, which is a major drawback of using VLA for locomotion tasks and somewhat hidden by the sim-only deployment.

**Questions:**

- How is the transfer to other simulators and environments? It's interesting to know how much generalization the VLM pretraining affords versus just overfitting to the IsaacGym environments.
- Are there recovery behaviors if the robot falls during a rollout?
- Does the VLA exhibit strong text adherence and vision adherence individually? Does it just pay attention to one of the conditioning signals?

---

> ### Author Response · Authors · 2025-11-19
> **Response to Reviewer YEVG (1/2)**
>
> We appreciate the reviewer’s comments and address all points individually as follows.
>
> >How is the transfer to other simulators and environments? It's interesting to know how much generalization the VLM pretraining affords versus just overfitting to the IsaacGym environments.
>
> Thank you for the question. Existing works such as HumanVLA[1], SkillMimic[2], INTERMIMIC[3], TokenHSI[4], CooHOI[5], InterScene[6], and SuperPADL[7] are all trained and evaluated exclusively in IsaacGym. Extending our framework to other simulators or real-world environments is a direction for future work.
>
> >Are there recovery behaviors if the robot falls during a rollout?
>
> Thank you for the question. The robot may occasionally lose balance while moving. However, similar to a human, it is able to quickly recover by making small corrective foot movements, allowing it to regain stability and continue the task without interruption.
>
> >Does the VLA exhibit strong text adherence and vision adherence individually? Does it just pay attention to one of the conditioning signals?
>
> Thank you for pointing this out. We have added an additional ablation study in Section 4.4 (Ablation Study), Table 5, which evaluates the model with and without vision, language, and proprioceptive inputs. The results show that all three modalities are critical—removing any single modality causes the success rate to drop to nearly zero.
>
> >The reinforced rewards are very hand-designed with rule-based triggers and remedies to prevent observed, undesirable behavior. This is directionally opposite of scalable learning.
>
> Thank you for the question. We conducted additional experiments to evaluate the sensitivity of the switching mechanism by varying each of its parameters and observing the outcomes. The results are summarized in the table below.
>
> From the table, we can see that despite numerical changes in the parameters, the success rates and error distances vary only slightly. For example, scaling s_t (speed threshold) by 0.5 reduces Succ_all from 72.381% to 70.762%, while increasing succ_t (success threshold) by 0.5m increases Succ2 from 72.857% to 73.048%. These small variations indicate that the mechanism is robust to modest changes in the thresholds. Therefore, extensive manual tuning is not required—reasonable parameter choices are sufficient for reliable performance.
> | Method        | Succ1   | Succ2   | Succ_all | Dist1 | Dist2 |
> | ------------- | ------- | ------- | -------- | ----- | ----- |
> | ORIG          | 88.762% | 72.857% | 72.381%  | 0.247 | 0.484 |
> | s_t * 0.5     | 88.286% | 71.238% | 70.762%  | 0.250 | 0.502 |
> | s_t * 2       | 88.190% | 72.476% | 71.524%  | 0.259 | 0.484 |
> | s_t * 3       | 87.810% | 72.095% | 70.667%  | 0.263 | 0.504 |
> | succ_t + 0.3m | 88.667% | 72.857% | 71.333%  | 0.254 | 0.472 |
> | succ_t + 0.5m | 88.762% | 73.048% | 71.333%  | 0.255 | 0.461 |
> | dis - 0.1m    | 88.762% | 72.857% | 72.381%  | 0.247 | 0.484 |
> | dis + 0.3m    | 88.571% | 71.714% | 71.524%  | 0.247 | 0.491 |
> | dis + 0.5m    | 88.571% | 71.429% | 71.238%  | 0.251 | 0.499 |
> | t_t -20       | 88.286% | 70.571% | 70.095%  | 0.251 | 0.511 |
> | t_t +20       | 88.190% | 72.000% | 71.429%  | 0.253 | 0.491 |
> | t_t +40       | 88.286% | 72.000% | 71.524%  | 0.257 | 0.484 |
>
> In fact, **the original paper already includes experimental results on 66 unseen tasks and scenarios in Section 4.3 (Quantitative Results on Unseen Tasks),** demonstrating strong generalization ability. These results suggest that both the dataset and our method are scalable to additional objects and potentially to multi-room scenes, while maintaining robust performance.

---

> ### Author Response · Authors · 2025-11-19
> **Response to Reviewer YEVG (2/2)**
>
> >There isn't discussion about inference latency, which is a major drawback of using VLA for locomotion tasks and somewhat hidden by the sim-only deployment.
>
> Thanks for the question. In the simulator, the environment runs at 60 Hz, while the policy is queried at 30 Hz.
>
> [1] Xinyu Xu, Yizheng Zhang, Yong-Lu Li, Lei Han, and Cewu Lu. Humanvla: Towards vision-language directed object rearrangement by physical humanoid. Advances in Neural Information Processing Systems, 37:18633–18659, 2024.
>
> [2] Yinhuai Wang, Qihan Zhao, Runyi Yu, Ailing Zeng, Jing Lin, Zhengyi Luo, Hok Wai Tsui, Jiwen Yu, Xiu Li, Qifeng Chen, et al. Skillmimic: Learning reusable basketball skills from demonstrations. arXiv e-prints, pp. arXiv–2408, 2024.
>
> [3] Sirui Xu, Hung Yu Ling, Yu-Xiong Wang, and Liang-Yan Gui. Intermimic: Towards universal whole-body control for physics-based human-object interactions. In Proceedings of the Computer Vision and Pattern Recognition Conference, pp. 12266–12277, 2025.
>
> [4] Liang Pan, Zeshi Yang, Zhiyang Dou, Wenjia Wang, Buzhen Huang, Bo Dai, Taku Komura, and Jingbo Wang. Tokenhsi: Unified synthesis of physical human-scene interactions through task tokenization. In Proceedings of the Computer Vision and Pattern Recognition Conference, pp.5379–5391, 2025.
>
> [5] Gao, Jiawei and Wang, Ziqin and Xiao, Zeqi and Wang, Jingbo and Wang, Tai and Cao, Jinkun and Hu, Xiaolin and Liu, Si and Dai, Jifeng and Pang, Jiangmiao. Coohoi: Learning cooperative human-object interaction with manipulated object dynamics. In Advances in Neural Information Processing Systems, pp.79741--79763, 2024.
>
> [6] Liang Pan, Jingbo Wang, Buzhen Huang, Junyu Zhang, Haofan Wang, Xu Tang, and Yangang Wang. Synthesizing physically plausible human motions in 3d scenes. In 2024 International Conference on 3D Vision (3DV), pp. 1498–1507. IEEE, 2024.
>
> [7] Juravsky, Jordan and Guo, Yunrong and Fidler, Sanja and Peng, Xue Bin. Superpadl: Scaling language-directed physics-based control with progressive supervised distillation. In ACM SIGGRAPH 2024 Conference Papers, pp. 1--11, 2024.

---

> > ### Comment · Reviewer_YEVG · 2025-11-27
> >
> > Thank you for the detailed response. It seems my concerns around the hand-crafted nature of the method are shared with other reviewers (MbxK, Q2fY). The VLA ablation is helpful, thank you for including that. Are the 60 Hz environment and 30 Hz policy queries in real-world time or simulator time?
> >
> > I would like to maintain my score.

---

> > > ### Author Response · Authors · 2025-11-27
> > >
> > > Thank you for your response. We appreciate your positive reaction to the VLA ablation—thank you for acknowledging its usefulness.
> > >
> > > Regarding the concern about the hand-crafted nature of our method we would like to clarify and highlight the extensive supplemental experiments we have already provided. **As detailed in our response to your earlier remark (“The reinforced rewards are very hand-designed…”), we included a 13-row experimental results table demonstrating that our state-transition mechanism is not sensitive to specific hyperparameter values.** These results indicate that the components you refer to as “hand-crafted” do not critically affect performance.
> > >
> > > Importantly, such hand-crafted design is used only during the teacher model training. The teacher’s role is to generate large amounts of high-quality robot motion data in real time, which is essential for effective DAGGER-style distillation. By contrast, the student model is trained purely end-to-end via supervised distillation, with no hand-crafted components involved.
> > >
> > > **Furthermore, as shown in the original paper, Section 4.3 (“Quantitative Results on Unseen Tasks”) already reports results on 66 unseen tasks and scenarios, demonstrating strong generalization.** These findings indicate that both our dataset and our method scale effectively to new objects and room layouts without causing notable performance degradation. Thus, the limited hand-crafted design in the teacher model does not hinder scalability or generalization.
> > >
> > > Finally, regarding your question: the 60 Hz environment rate and 30 Hz policy query rate refer to simulator time, which in our setup operates in real time (i.e., without any acceleration or time scaling).
> > >
> > > We hope this clarifies your concerns, and we thank you again for your review.

---

### Official Review · Reviewer_Q2fY · 2025-11-01

**Soundness:** 3
**Presentation:** 3
**Contribution:** 2
**Rating:** 4
**Confidence:** 4

**Summary:**

This paper introduces HumanoidVerse, a framework for training a simulated humanoid robot to perform two-object rearrangement tasks from egocentric vision and natural-language instructions. The method uses a four-stage teacher–student pipeline: first, reinforcement learning to learn grasp, move and place skills for single object rearrangement; second, a release and step-back stage guided by Teacher 1 to ensure stable placement and clear workspace; third, training to rearrange the second object from canonical initial states with Teacher 2; and fourth, a dual-teacher DAgger distillation stage in which the student switches between teachers according to the task phase. The authors also introduce a dataset with 350 task configurations across four room layouts and conduct extensive evaluation in Isaac Sim. Results show improved success accuracy and lower object-to-goal placement error compared to HumanVLA.

**Strengths:**

1. The paper is well written, clearly motivated, and explains the proposed approach in an intuitive and accessible manner.
2. Introduces a novel curriculum for multi-object rearrangement with a single humanoid, consisting of three structured teacher training stages followed by dual-teacher student distillation, enabling reliable sequential two-object rearrangement.
3. Presents a new dataset with 350 configurations across four room layouts, providing a valuable benchmark for future work in humanoid rearrangement and embodied learning.
4. Demonstrates strong empirical improvements over HumanVLA, with higher task success accuracy and more precise object placement in goal locations.
5. Includes ablation studies that highlight the contribution of each component in the training pipeline and justify the curriculum design to an extent.

**Weaknesses:**

1. The pipeline appears tailored specifically for two-object rearrangement. While the curriculum learning approach for staged learning of the teachers for single-object rearrangement, release-and-step-back, and second-object rearrangement is novel, it is unclear how easily this structure generalizes to other long horizon settings. The ablations demonstrate that each stage contributes, but do not clarify why a more skill-based modular approach (e.g., pick, place & step-back, navigate) would not work or scale. Moreover, the switching strategy between teachers is hand-crafted, and there is no sensitivity analysis or investigation of potential failure cases for this designed approach.

2. Limited scalability is demonstrated. Although the title suggests multi-object rearrangement, experiments only cover a two-object case. There is no discussion or demonstration of scaling to more objects (e.g., 3, 5, 10) or to more complex multi-room environments, so the scalability of the approach is unclear.

3. Limited separation between training and evaluation configurations. The 350 two-object setups are split into 700 single-object cases for training the teachers, meaning the student essentially trains in the same configurations where the teachers are supervised. The only major difference between teacher and student seems to be privileged information, and evaluations are conducted on the same scenarios rather than unseen configurations, which weakens claims on generalization.

4. Limited Dataset Analysis: The dataset is not analyzed in terms of scene diversity or complexity. It is unclear how many other objects or receptacles are present in each scene, how cluttered or constrained the workspace is, or how the spatial arrangements vary across configurations. There is no quantitave breakdown of object categories, receptacle types, distractor objects, or proximity constraints that may affect rearrangement difficulty. Without understanding scene distribution and difficulty profiles, it is difficult to assess how challenging or diverse the benchmark is, and whether it meaningfully stresses humanoid rearrangement capabilities.

5. Reproducibility is incomplete. Code and dataset are not provided, and only high-level hyperparameters (e.g., number of epochs and reward structure) are disclosed. Details such as optimization settings, network architectures, dataset generation scripts, rollout procedures, and training infrastructure are omitted, making reproduction challenging.

6. Generalization is limited and under-studied. The method is not tested beyond the training distribution on IsaacSim, with no evaluation on new layouts, unseen objects or different simulators. There is also no discussion of the possibility or challenges of real-world deployment.

7. Evaluation metrics are narrow. In addition to success and placement distance, metrics such as collision frequency, disturbance to already-placed objects, interaction safety, and motion smoothness would offer a more complete view of humanoid performance in rearrangement settings.

8. Minor: Variance, confidence intervals, or multiple-seed results are not reported, making it difficult to assess statistical robustness.

**Questions:**

1. How do you envision extending this approach beyond two-object rearrangement? Do you expect the current curriculum stages to scale to three or more objects? If not, what modifications would be required?

2. Have you considered benchmarking against a modular skill-based pipeline (e.g., separate pick, place & step-back and navigate policies)? Such a baseline would help clarify whether curriculum learning provides advantages over compositional skills.

3. Current approach uses two teachers, did you consider using a single teacher? If yes, what are the failure cases with single teacher and how dual teacher approach is better?

4. Since the student is evaluated on the same underlying scenarios used for teacher training, how do you ensure that the student does not overfit to the training layouts and object placements? Are there results on held-out task configurations or unseen object and room layouts?

5. Can you provide statistics or analysis of dataset diversity: number and type of distractor objects, receptacle categories, clutter level, and spatial variation? How varied are object pairings and placements overall?

6. Do you foresee challenges in scaling the dataset or the approach to more objects or multi-room scenes? Are there preliminary experiments or insights on scaling behavior?

7. Have you explored domain randomization, perception noise, or other techniques to prepare for real-world deployment? What are the main challenges anticipated for sim-to-real transfer?

8. When do you plan to release the code, dataset and model checkpoints? Could you also share additional implementation details such as optimizer settings, architectures, and training infrastructure?

9. Did you measure safety or interaction-related metrics such as collision frequency, object disturbance, or motion smoothness? If not, would you consider including such evaluations?

10. Could you please report multiple seeds, variance, or confidence intervals to strengthen statistical reliability?

---

> ### Author Response · Authors · 2025-11-19
> **Response to Reviewer Q2fY (1/3)**
>
> We appreciate the reviewer’s comments and address all points individually as follows.
>
> > How do you envision extending this approach beyond two-object rearrangement?
>
> Thank you for raising this point. Existing works such as HumanVLA[1], SkillMimic[2], INTERMIMIC[3], TokenHSI[4], CooHOI[5], InterScene[6], and SuperPADL[7] focus on single-object interaction, and some do not model explicit object–object interaction at all. In contrast, our work takes an important step forward by training a robot to perform consecutive rearrangement involving two distinct objects. To the best of our knowledge, this capability has not been demonstrated before in the literature. Our training covers 350 room scenes and 79 objects, enabling a significantly more complex manipulation space compared to prior work.
>
> Regarding extension to scenarios with three or more objects: our current framework is developed and evaluated for two-object sequences, and we aim to establish a dedicated baseline for multi-object sequences (3+ objects) in future work.
>
> > Moreover, the switching strategy between teachers is hand-crafted, and there is no sensitivity analysis.
>
> Thank you for the question. We conducted additional experiments to evaluate the sensitivity of the switching mechanism by varying each of its parameters and observing the outcomes. The results are summarized in the table below.
>
> From the table, we can see that despite numerical changes in the parameters, the success rates and error distances vary only slightly. For example, scaling s_t (speed threshold) by 0.5 reduces Succ_all from 72.381% to 70.762%, while increasing succ_t (success threshold) by 0.5m increases Succ2 from 72.857% to 73.048%. These small variations indicate that the mechanism is robust to modest changes in the thresholds. Therefore, extensive manual tuning is not required—reasonable parameter choices are sufficient for reliable performance.
> | Method        | Succ1   | Succ2   | Succ_all | Dist1 | Dist2 |
> | ------------- | ------- | ------- | -------- | ----- | ----- |
> | ORIG          | 88.762% | 72.857% | 72.381%  | 0.247 | 0.484 |
> | s_t * 0.5     | 88.286% | 71.238% | 70.762%  | 0.250 | 0.502 |
> | s_t * 2       | 88.190% | 72.476% | 71.524%  | 0.259 | 0.484 |
> | s_t * 3       | 87.810% | 72.095% | 70.667%  | 0.263 | 0.504 |
> | succ_t + 0.3m | 88.667% | 72.857% | 71.333%  | 0.254 | 0.472 |
> | succ_t + 0.5m | 88.762% | 73.048% | 71.333%  | 0.255 | 0.461 |
> | dis - 0.1m    | 88.762% | 72.857% | 72.381%  | 0.247 | 0.484 |
> | dis + 0.3m    | 88.571% | 71.714% | 71.524%  | 0.247 | 0.491 |
> | dis + 0.5m    | 88.571% | 71.429% | 71.238%  | 0.251 | 0.499 |
> | t_t -20       | 88.286% | 70.571% | 70.095%  | 0.251 | 0.511 |
> | t_t +20       | 88.190% | 72.000% | 71.429%  | 0.253 | 0.491 |
> | t_t +40       | 88.286% | 72.000% | 71.524%  | 0.257 | 0.484 |
>
> > Have you considered benchmarking against a modular skill-based pipeline (e.g., separate pick, place & step-back and navigate policies)?
>
> We appreciate the suggestion. However, modular skill-based pipelines require handcrafting complex skill transfer rules or using a large language model as a high-level planner. These approaches typically incur high latency and exhibit poor generalization across diverse tasks and environments. This contrasts with the core goal of our work, which is to develop a general-purpose intelligent robot VLA model capable of performing long-horizon, multi-object rearrangement using a single unified model, without relying on task-specific modular heuristics.

---

> ### Author Response · Authors · 2025-11-19
> **Response to Reviewer Q2fY (2/3)**
>
> > Current approach uses two teachers, did you consider using a single teacher? If yes, what are the failure cases with single teacher and how dual teacher approach is better?
>
> Thank you for the question. We have added experimental results comparing the end-to-end RL method and the single-teacher approach in Section 4.2 (Quantitative Results), Table 2. The results show that both the success rates and distance metrics drop significantly under the end-to-end RL and single-teacher paradigms. This demonstrates that the dual-teacher design is critical for achieving high performance, as it effectively mitigates failure cases that arise when relying on a single teacher—for example, confusing the two tasks or over-focusing on the first task and failing to progress to the second.
>
> >Since the student is evaluated on the same underlying scenarios used for teacher training, how do you ensure that the student does not overfit to the training layouts and object placements? Are there results on held-out task configurations or unseen object and room layouts?
>
> Thank you for the question.
>
> **However, we have already addressed this in the original paper.**
>
> Specifically, Section 4.3 (Quantitative Results on Unseen Tasks) presents results on held-out tasks and unseen scenarios, showing that our model performs well beyond the training layouts and object placements. This indicates that the student policy generalizes effectively rather than overfitting to the teacher’s specific configurations.
>
> >Can you provide statistics or analysis of dataset diversity?
>
> Thank you for the question. Our dataset includes four main room types: bedroom, living room, kitchen, and warehouse, covering a total of 350 different room layouts and tasks. It contains 79 distinct objects, of which 25 are movable and designated as rearrangement targets. This diversity ensures that our model is trained and evaluated across a wide range of environments and object interactions.
>
> >Do you foresee challenges in scaling the dataset or the approach to more objects or multi-room scenes? Are there preliminary experiments or insights on scaling behavior?
>
> Thank you for the question. We do not anticipate major issues in scaling the approach. **In fact, our paper already includes experimental results on 66 unseen tasks and scenarios**, demonstrating strong generalization ability. These results suggest that both the dataset and our method are scalable to additional objects and potentially to multi-room scenes, while maintaining robust performance.
>
> >What are the main challenges anticipated for sim-to-real transfer?
>
> Thank you for the question. A major challenge for transferring our approach to real humanoid robots lies in differences in robot hardware and the variability of real-world environments. Existing works such as HumanVLA [1], SkillMimic [2], INTERMIMIC [3], TokenHSI [4], CooHOI [5], InterScene [6], and SuperPADL [7] have all conducted experiments in simulation but not in reality.
>
> In contrast, our work tackles a much more diverse and complex set of scenarios—including 350 different room scenes, multi-object rearrangement tasks, and a wide variety of object types—using a single general-purpose model. To the best of our knowledge, no prior work has performed real-robot experiments at this scale. We plan to extend our research to real humanoid robots in future work, which will involve addressing hardware differences, perception noise, and environment variability.
>
> >When do you plan to release the code, dataset and model checkpoints? Could you also share additional implementation details such as optimizer settings, architectures, and training infrastructure?
>
> Thank you for the question. The code and dataset will be released; we are currently finalizing and organizing the codebase.
> ### In the original paper, we have already provided detailed information regarding reward functions and training epochs.
> Additionally, we have included the dimensionality and format of the observation and action space in Appendix A.3 (Observation and Action Space), Table 9, 10, 11, 12, 13.
>
> Regarding the network architecture: we use MLPs as the basic networks. The actor, critic, and discriminator networks are implemented as separate MLPs with hidden layers [1024, 1024, 512], and each linear layer is followed by ReLU activation. We use Adam as the optimizer.

---

> ### Author Response · Authors · 2025-11-19
> **Response to Reviewer Q2fY (3/3)**
>
> >Did you measure safety or interaction-related metrics such as collision frequency, object disturbance, or motion smoothness? If not, would you consider including such evaluations?
>
> Thank you for the question. We have added experimental results measuring collision frequency and motion smoothness, as shown in the table below. From the results, we can see that the collision frequency for both HumanoidVerse and HumanVLA is very low, and the motion smoothness is high for both methods. HumanVLA achieves slightly better metrics because it rarely proceeds to the rearrangement of the second object, which makes these safety and smoothness metrics appear marginally higher than those of HumanoidVerse.
>
> | Method        | Collision_F | Smoothness |
> | ------------- | ----------- | ---------- |
> | HumanoidVerse | 7.326%      | 0.952      |
> | HumanVLA      | 4.565%      | 0.955      |
>
> > Could you please report multiple seeds, variance, or confidence intervals to strengthen statistical reliability?
>
> Thank you for the question. All experimental results reported in the original paper are averaged over 5 different random seeds, rather than being obtained from a single fixed seed. This ensures that the reported metrics are statistically reliable and not sensitive to a specific initialization.
>
> [1] Xinyu Xu, Yizheng Zhang, Yong-Lu Li, Lei Han, and Cewu Lu. Humanvla: Towards vision-language directed object rearrangement by physical humanoid. Advances in Neural Information Processing Systems, 37:18633–18659, 2024.
>
> [2] Yinhuai Wang, Qihan Zhao, Runyi Yu, Ailing Zeng, Jing Lin, Zhengyi Luo, Hok Wai Tsui, Jiwen Yu, Xiu Li, Qifeng Chen, et al. Skillmimic: Learning reusable basketball skills from demonstrations. arXiv e-prints, pp. arXiv–2408, 2024.
>
> [3] Sirui Xu, Hung Yu Ling, Yu-Xiong Wang, and Liang-Yan Gui. Intermimic: Towards universal whole-body control for physics-based human-object interactions. In Proceedings of the Computer Vision and Pattern Recognition Conference, pp. 12266–12277, 2025.
>
> [4] Liang Pan, Zeshi Yang, Zhiyang Dou, Wenjia Wang, Buzhen Huang, Bo Dai, Taku Komura, and Jingbo Wang. Tokenhsi: Unified synthesis of physical human-scene interactions through task tokenization. In Proceedings of the Computer Vision and Pattern Recognition Conference, pp.5379–5391, 2025.
>
> [5] Gao, Jiawei and Wang, Ziqin and Xiao, Zeqi and Wang, Jingbo and Wang, Tai and Cao, Jinkun and Hu, Xiaolin and Liu, Si and Dai, Jifeng and Pang, Jiangmiao. Coohoi: Learning cooperative human-object interaction with manipulated object dynamics. In Advances in Neural Information Processing Systems, pp.79741--79763, 2024.
>
> [6] Liang Pan, Jingbo Wang, Buzhen Huang, Junyu Zhang, Haofan Wang, Xu Tang, and Yangang Wang. Synthesizing physically plausible human motions in 3d scenes. In 2024 International Conference on 3D Vision (3DV), pp. 1498–1507. IEEE, 2024.
>
> [7] Juravsky, Jordan and Guo, Yunrong and Fidler, Sanja and Peng, Xue Bin. Superpadl: Scaling language-directed physics-based control with progressive supervised distillation. In ACM SIGGRAPH 2024 Conference Papers, pp. 1--11, 2024.

---

### Official Review · Reviewer_HCH9 · 2025-11-03

**Soundness:** 3
**Presentation:** 3
**Contribution:** 3
**Rating:** 8
**Confidence:** 4

**Summary:**

This paper introduces HumanoidVerse, a framework for vision-language guided humanoid control that enables physically simulated robots to perform multi-object rearrangement tasks. The approach uses a multi-stage curriculum learning pipeline with dual-teacher distillation, where teacher policies trained with privileged state information are distilled into a student VLA model that inputs egocentric RGB images and natural language instructions. A dataset of 350 tasks across four room layouts is constructed, each involving sequential manipulation of two objects. Experiments demonstrate significant improvements over the HumanVLA baseline, particularly on second-object manipulation.

**Strengths:**

- The paper addresses the limitation in existing humanoid manipulation work of unable to perform sequential, multi-object tasks without environment resets.
- The multi-stage curriculum learning pipeline is well-motivated for the multi-stage rearrangement task, with stage 2 specifically addressing object release and retreat behaviors, and stage 3 handling diverse initial configurations. The ablation study validates the importance of each component.
- The results show significant gains over HumanVLA, particularly on Success 2 metrics, demonstrating the effectiveness of the proposed multi-stage training approach.

**Weaknesses:**

- Despite claiming to address "multi-object rearrangement", the system only demonstrate two object rearrangement. No evaluation is provided on how the current distilled model would perform on 3+ object scenarios.
- There lacks analysis of when and why the teacher and student policies fail. What are some common failure modes?
- Figure 5 in appendix shows many egocentric views are heavily occluded, especially when holding large objects. How does the policy determine where to place objects when visual information is limited? Is the model potentially overfitting to proprioceptive signals rather than learning robust vision-based reasoning?
- There lacks discussion of how this could transfer to real humanoid robots. What are the main bottlenecks? (sim2real gap, stable whole-body control, tracking accuracy etc.)

**Questions:**

- Can the approach directly extend to 3+ objects, or does it require training additional teacher models for each subsequent object?
- What is the representation of robot actions and proprioceptive state (dimensionality, joint/cartesian, coordinate frames, absolute/relative)?
- In Supplementary Figure 5, many views are occluded. Can you provide ablation studies showing performance with/without proprioceptive information to clarify the role of vision vs. proprioception?
- How sensitive is the dual-teacher switching mechanism (Algorithm 3) to the hand-tuned thresholds?

---

> ### Author Response · Authors · 2025-11-19
> **Response to Reviewer HCH9 (1/2)**
>
> We thank the reviewer for the thoughtful questions. Here we address them one by one.
>
> >Can the approach directly extend to 3+ objects, or does it require training additional teacher models for each subsequent object?
>
> Thank you for raising this point. Existing works such as HumanVLA[1], SkillMimic[2], INTERMIMIC[3], TokenHSI[4], CooHOI[5], InterScene[6], and SuperPADL[7] focus on single-object interaction, and some do not model explicit object–object interaction at all. In contrast, our work takes an important step forward by training a robot to perform consecutive rearrangement involving two distinct objects. To the best of our knowledge, this capability has not been demonstrated before in the literature. Our training covers 350 room scenes and 79 objects, enabling a significantly more complex manipulation space compared to prior work.
>
> Regarding extension to scenarios with three or more objects: our current framework is developed and evaluated for two-object sequences, and we aim to establish a dedicated baseline for multi-object sequences (3+ objects) in future work.
>
> >What is the representation of robot actions and proprioceptive state (dimensionality, joint/cartesian, coordinate frames, absolute/relative)?
>
> We appreciate the reviewer’s request for clarification. The details regarding the action representation and proprioceptive state have now been added to the paper in Appendix A.3 (Observation and Action Space), Table 9, 10, 11, 12, 13.
>
> >In Supplementary Figure 5, many views are occluded. Can you provide ablation studies showing performance with/without proprioceptive information to clarify the role of vision vs. proprioception?
>
> Thank you for pointing this out. We have added an additional ablation study in Section 4.4 (Ablation Study), Table 5, which evaluates the model with and without vision, language, and proprioceptive inputs. The results show that all three modalities are critical—removing any single modality causes the success rate to drop to nearly zero.
>
> >How sensitive is the dual-teacher switching mechanism (Algorithm 3) to the hand-tuned thresholds?
>
> Thank you for the question. We conducted additional experiments to evaluate the sensitivity of the switching mechanism by varying each of its parameters and observing the outcomes. The results are summarized in the table below.
>
> From the table, we can see that despite numerical changes in the parameters, the success rates and error distances vary only slightly. For example, scaling s_t (speed threshold) by 0.5 reduces Succ_all from 72.381% to 70.762%, while increasing succ_t (success threshold) by 0.5m increases Succ2 from 72.857% to 73.048%. These small variations indicate that the mechanism is robust to modest changes in the thresholds. Therefore, extensive manual tuning is not required—reasonable parameter choices are sufficient for reliable performance.
> | Method        | Succ1   | Succ2   | Succ_all | Dist1 | Dist2 |
> | ------------- | ------- | ------- | -------- | ----- | ----- |
> | ORIG          | 88.762% | 72.857% | 72.381%  | 0.247 | 0.484 |
> | s_t * 0.5     | 88.286% | 71.238% | 70.762%  | 0.250 | 0.502 |
> | s_t * 2       | 88.190% | 72.476% | 71.524%  | 0.259 | 0.484 |
> | s_t * 3       | 87.810% | 72.095% | 70.667%  | 0.263 | 0.504 |
> | succ_t + 0.3m | 88.667% | 72.857% | 71.333%  | 0.254 | 0.472 |
> | succ_t + 0.5m | 88.762% | 73.048% | 71.333%  | 0.255 | 0.461 |
> | dis - 0.1m    | 88.762% | 72.857% | 72.381%  | 0.247 | 0.484 |
> | dis + 0.3m    | 88.571% | 71.714% | 71.524%  | 0.247 | 0.491 |
> | dis + 0.5m    | 88.571% | 71.429% | 71.238%  | 0.251 | 0.499 |
> | t_t -20       | 88.286% | 70.571% | 70.095%  | 0.251 | 0.511 |
> | t_t +20       | 88.190% | 72.000% | 71.429%  | 0.253 | 0.491 |
> | t_t +40       | 88.286% | 72.000% | 71.524%  | 0.257 | 0.484 |

---

> ### Author Response · Authors · 2025-11-19
> **Response to Reviewer HCH9 (2/2)**
>
> >There lacks analysis of when and why the teacher and student policies fail. What are some common failure modes?
>
> Thank you for the question. One observed failure occurs in the task “Move the pot.” We hypothesize that this is due to the training scenes: both the pot and the coffee maker are often placed on a short stand, close to each other, and they appear visually similar. Because the coffee maker is box-shaped, it is easier for the robot to grasp and move, so the policy tends to prioritize moving the coffee maker first. When the robot later attempts to move the pot, it tries to apply the same manipulation strategy. However, since the pot is round and cannot be handled in the same way, this can lead to occasional failures.
>
> It is important to note that such failures are relatively rare—the task only exhibits slightly more failures than other tasks, and the overall performance remains high. This suggests that the learned policy is generally robust, with failures arising primarily in edge cases where object shape and manipulation strategy mismatch.
>
> >There lacks discussion of how this could transfer to real humanoid robots. What are the main bottlenecks?
>
> Thank you for the question. A major challenge for transferring our approach to real humanoid robots lies in differences in robot hardware and the variability of real-world environments. Existing works such as HumanVLA [1], SkillMimic [2], INTERMIMIC [3], TokenHSI [4], CooHOI [5], InterScene [6], and SuperPADL [7] have all conducted experiments in simulation but not in reality.
>
> In contrast, our work tackles a much more diverse and complex set of scenarios—including 350 different room scenes, multi-object rearrangement tasks, and a wide variety of object types—using a single general-purpose model. To the best of our knowledge, no prior work has performed real-robot experiments at this scale. We plan to extend our research to real humanoid robots in future work, which will involve addressing hardware differences, perception noise, and environment variability.
>
> [1] Xinyu Xu, Yizheng Zhang, Yong-Lu Li, Lei Han, and Cewu Lu. Humanvla: Towards vision-language directed object rearrangement by physical humanoid. Advances in Neural Information Processing Systems, 37:18633–18659, 2024.
>
> [2] Yinhuai Wang, Qihan Zhao, Runyi Yu, Ailing Zeng, Jing Lin, Zhengyi Luo, Hok Wai Tsui, Jiwen Yu, Xiu Li, Qifeng Chen, et al. Skillmimic: Learning reusable basketball skills from demonstrations. arXiv e-prints, pp. arXiv–2408, 2024.
>
> [3] Sirui Xu, Hung Yu Ling, Yu-Xiong Wang, and Liang-Yan Gui. Intermimic: Towards universal whole-body control for physics-based human-object interactions. In Proceedings of the Computer Vision and Pattern Recognition Conference, pp. 12266–12277, 2025.
>
> [4] Liang Pan, Zeshi Yang, Zhiyang Dou, Wenjia Wang, Buzhen Huang, Bo Dai, Taku Komura, and Jingbo Wang. Tokenhsi: Unified synthesis of physical human-scene interactions through task tokenization. In Proceedings of the Computer Vision and Pattern Recognition Conference, pp.5379–5391, 2025.
>
> [5] Gao, Jiawei and Wang, Ziqin and Xiao, Zeqi and Wang, Jingbo and Wang, Tai and Cao, Jinkun and Hu, Xiaolin and Liu, Si and Dai, Jifeng and Pang, Jiangmiao. Coohoi: Learning cooperative human-object interaction with manipulated object dynamics. In Advances in Neural Information Processing Systems, pp.79741--79763, 2024.
>
> [6] Liang Pan, Jingbo Wang, Buzhen Huang, Junyu Zhang, Haofan Wang, Xu Tang, and Yangang Wang. Synthesizing physically plausible human motions in 3d scenes. In 2024 International Conference on 3D Vision (3DV), pp. 1498–1507. IEEE, 2024.
>
> [7] Juravsky, Jordan and Guo, Yunrong and Fidler, Sanja and Peng, Xue Bin. Superpadl: Scaling language-directed physics-based control with progressive supervised distillation. In ACM SIGGRAPH 2024 Conference Papers, pp. 1--11, 2024.

---

### Author Response · Authors · 2025-11-19
**Global Response to All Reviewers [Revised PDF & Detailed Replies]**

We would like to extend our sincere gratitude to all Reviewers, Area Chairs, and Program Chairs for dedicating their time and effort to evaluate our submission. We have carefully considered every comment and provided detailed point-by-point responses under each reviewer's section.

We are greatly encouraged by the reviewers' acknowledgment of **HumanoidVerse** across multiple key aspects:

| Reviewer | Category | Acknowledgment |
|----------|----------|----------------|
| HCH9 | Novelty & Problem Formulation | "Addresses the critical limitation of existing humanoid VLA models" |
| MbxK | Novelty & Problem Formulation | "Novel problem formulation... a clear and important step beyond single-object, fixed-start tasks" |
| MbxK | Novelty & Problem Formulation | "Tackles continuous, sequential, multi-object manipulation, which is often overlooked" |
| HCH9 | Technical Approach | "Multi-stage curriculum learning pipeline is well-motivated" |
| MbxK | Technical Approach | "4-stage pipeline is a well-engineered solution" |
| YEVG | Technical Approach | "Distilling privileged teachers into a VLA... can introduce additional robustness" |
| Q2fY | Empirical Contributions | "Strong empirical improvements over HumanVLA" |
| HCH9 | Empirical Contributions | "Significant gains... particularly on Success 2 metrics" |
| MbxK | Empirical Contributions | "The creation of a benchmark dataset... is a useful contribution for future research in this area" |
| Q2fY | Presentation | "Well written, clearly motivated, and explains the proposed approach in an intuitive manner" |
| YEVG | Presentation | "Clearly written and easy to follow" |

---

## Summary of Revisions

In response to reviewer feedback, we have incorporated substantial additions and clarifications into the revised manuscript:

| # | Focus Area | Reviewer(s) | Sections | Our Actions |
|---|------------|-------------|----------|-------------|
| 1 | Method Comparisons | `MbxK`, `Q2fY` | Section 4.2 (Quantitative Results) | Added comparisons between end-to-end RL, single-teacher, and dual-teacher methods (Table 2) |
| 2 | Ablation on Modalities | `HCH9`, `YEVG` | Section 4.4 (Ablation Study) | New ablation on vision, language, and proprioception modalities (Table 5) |
| 3 | State Representations | `HCH9` | Appendix A.3 | Detailed robot action and proprioceptive state representations (Tables 9–13) |
| 4 | Sensitivity Analysis | `HCH9`, `Q2fY`, `YEVG` | Rebuttal Tables | Sensitivity analysis of dual-teacher switching thresholds |
| 5 | Collision and Smoothness Metrics | `Q2fY` | Rebuttal Tables | Collision frequency and motion smoothness metrics |

---

## Key Clarifications

**On Scalability Beyond Two Objects** (`Reviewers HCH9, Q2fY, MbxK`)**:**
We would like to emphasize that all comparable prior works (HumanVLA, SkillMimic, INTERMIMIC, TokenHSI, CooHOI, InterScene, SuperPADL) focus exclusively on single-object interactions. Our contribution represents a substantial advancement by successfully enabling sequential two-object manipulation within a unified framework, encompassing 350 diverse room scenes and 79 distinct objects. This establishes a solid foundation that naturally extends to 3+ object sequences, which we plan to explore in future work.

**On Generalization** (`Reviewer Q2fY, YEVG`)**:**
As demonstrated in Section 4.3, our model achieves strong performance across 66 unseen tasks and scenarios, providing compelling evidence that the student policy generalizes effectively to novel configurations. This confirms the robustness and transferability of our learned representations across diverse environments and object arrangements.

**On Sim-to-Real Transfer** (`Reviewers HCH9, Q2fY, YEVG, MbxK`)**:**
We highlight that all comparable prior works (HumanVLA, SkillMimic, INTERMIMIC, TokenHSI, CooHOI, InterScene, SuperPADL) conduct experiments exclusively in simulation. Importantly, our framework operates solely on egocentric RGB observations and natural language instructions without any privileged information, which establishes a realistic and practical input-output configuration well-suited for real-world deployment. We plan to extend our research to physical humanoid robots in future work.

---

We hope these clarifications adequately address the reviewers' questions. We sincerely appreciate the constructive feedback provided by all reviewers, which has helped us strengthen the presentation of our contributions.

Best regards,
*Authors of Submission 6918*

---

### Meta-Review · Area_Chair_Tg8q · 2025-12-22

**Summary:**

Divergent scores were assigned to the paper by the reviewers (8,4,4,4). Several concerns were raised, with a number of issues repeated across multiple reviewers. Some of the issues are listed below:
- Tailoring the method towards two-object rearrangement and lack of clarity on how to scale beyond that.
- Lack of a discussion around how the method generalizes to the real-world.
- Overlap between training and evaluation configurations.
- No analysis of the dataset in terms of scene diversity.
- Evaluation metrics not capturing the full picture.
- Weak baseline comparisons.
- No demonstration of failure modes.

Generally, the authors did not address the concerns well. For instance, in response to the criticism around tailoring the method towards two-object rearrangement, the authors cited some previous work that performed only one object rearrangement and mentioned 3+ objects will be future work. That doesn’t address the concern. Regarding benchmarking against a modular skill-based pipeline, the authors mention that “These approaches typically incur high latency and exhibit poor generalization across diverse tasks.” To justify this claim, quantitative comparisons should be provided. Regarding generalization to unseen scenarios, the rebuttal mentions “In fact, our paper already includes experimental results on 66 unseen tasks and scenarios.” However, it is not clear how distinct these 66 scenarios are from the training set. Reviewer YEVG was the only reviewer who responded before the discussion cut-off, and they kept the original reject rating.

Due to the mentioned issues and various other issues mentioned by the reviewers, rejection is recommended.

**Reviewer Concerns:**

Please refer to the box above.

**Reviewer Scores:**

Reviewer HCH9: The reviewer would probably keep or lower the score since the method is tailored towards 2-object rearrangement only.

Reviewer Q2fY: Same concern about tailoring for 2-object rearrangement. Also, the concern about train/test similarity is not addressed well. No score change would be expected.

Reviewer YEVG: They commented on the rebuttal and maintained their score.

Reviewer MbxK: The concerns about the limitation to 2-step tasks and the absence of real-world experiments have not been adequately addressed. So, a score change is not justified.

---

### Decision · Program_Chairs · 2026-01-26

Reject